# Induction of cardiac fibulin-4 protects against pressure overload-induced cardiac hypertrophy and heart failure
E. D. van Deel[1,2,14], M. Snelders [1,14], N. van Vliet[1], L. te Riet[3,4], T. P. P. van den Bosch[5], L. R. Fiedler[6], A. C. C. van Spreeuwel[7], N. A. M. Bax[7], N. Boontje[8], C. M. Halabi [9], T. Sasaki [10], D. P. Reinhardt [11,12], J. van der Velden[8], C.V.C. Bouten [7], J. H. von der Thüsen [5], A. H. J. Danser[3], D. J. Duncker [2], M. D. Schneider [6], I. van der Pluijm [1,4] & J. Essers [1,4,13] ✉

The prevailing view of fibulin-4 deficient mice is that the cardiac phenotype is the result of aortic and/or valvular disease. In the present study, we have tested whether the cardiac phenotype is, at least in part, the consequence of primary cardiac effects of fibulin-4. We have found fibulin-4 expression to be activated throughout the myocardium in wildtype (fibulin-4$^{+/+}$) C57Bl/6J;129 Sv mice subjected to transverse aortic constriction (TAC). In contrast, haploinsufficient fibulin-4$^{+/R}$ mice exposed to severe TAC do not show this increase in myocardial fibulin-4 expression, but display altered physical properties of myocardial tissue. Moreover, TAC-induced cardiac fibrosis, pulmonary congestion, and mortality are aggravated in fibulin-4$^{+/R}$ mice. In vitro investigations of myocardial tissue show that fibulin-4 deficiency results in cardiomyocyte hypertrophy, and a decreased beating frequency and contractile force. In conclusion, we demonstrate functions for fibulin-4 in cardiac homeostasis and show that reduced fibulin-4 expression drives myocardial disease in response to cardiac pressure overload, independent of aortic valvular pathology.

The integrity of elastic fibers is crucial for cardiovascular homeostasis and postnatal survival. This is illustrated by severe cardiovascular complications and premature death in patients and mice with deficiencies in elastic fiber building blocks. Elastic fibers are essential extracellular matrix (ECM) macromolecules comprised of an elastin core surrounded by a mantle of fibrillin-rich microfibrils. The extracellular matrix (ECM) provides structural support to cells, but its composition and mechanical properties are also essential for cell development, signaling and function[1]. The process of elastic fiber formation requires orchestrated deposition of critical elastic fiber components in a hierarchical order. Several ECM-related disorders frequently exhibit cardiovascular manifestations, including thoracic aortic aneurysm, aortic valve regurgitation and cardiac remodeling and dysfunction, which are generally attributed to the aortic valve abnormalities in these patients[2,3].

Connective tissue disorders, such as Marfan syndrome resulting from mutations in the ECM protein fibrillin 1, have demonstrated a direct impact on the heart under conditions of cardiac pressure overload. Fibrillin 1-deficient mice show dilated cardiomyopathy directly related to the reduced expression of fibrillin 1, accompanied by increased expression of angiotensin II type 1 receptor (AT1R) and reduced focal adhesion kinase (FAK) activity[4]. FAK plays a critical role in cardiac development,

remodeling, and signaling pathways involved in heart function and response to stress[5]. Fibulins are important components that facilitate protein-protein interactions in the ECM. The fibulin family consists of 3 short and 4 long fibulins. Fibulin-4 and -5 are long fibulins and have been implicated in elastic fiber assembly and arterial and vascular elastogenesis. Fibulin-4 contributes to the formation of elastic fibers and is indispensable for proper ECM assembly[6,7]. Fibulin-4 regulates collagen cross-linking and plays a role in lysyl oxidase activation[8,9]. Mutations within the fibulin-4 gene in humans result in cutis laxa syndrome, a condition marked not only by loose skin but also by cardiovascular pathology similar to Marfan syndrome[10–14]. We previously demonstrated that mice with a systemic 75% reduction of fibulin-4 expression (fibulin-4$^{R/R}$) share key features with the human disease phenotype, including aortic aneurysm formation, aortic valve disease, increased transforming growth factor beta (TGF-β) signaling, and impaired cardiac function[15,16]. In contrast, fibulin-4$^{+/R}$ mice, with a milder 50% reduction in fibulin-4 expression, develop no apparent cardiovascular abnormalities[15,16]. Although fibulin-4 is linked to aortic valve disease, the underlying cause for cardiac failure in fibulin-4 deficiency remains incompletely understood.

Several observations indicate that fibulin-4 reduction has a direct cardiac pathological effect, in addition to valvular regurgitation. First,

cardiac pathology was much more pronounced in fibulin-4[R/R] mice compared to other models of aortic regurgitation[17–20]. Second, the ubiquitous presence of fibulin-4 in cardiovascular tissues, including the interstitial space surrounding cardiomyocytes, further suggests fibulin-4 to have a primary cardiac function apart from vasculo-valvular integrity[21]. Third, fibulin-4 deficiency was previously linked to abnormal mechanosensing[22], and thus might be involved in AT1R signaling, similar to fibrillin-1 in Marfan syndrome[4]. Lastly, in the pressure-overloaded heart, myofibroblasts actively deposit collagen at areas of injury. Because fibulin-4 is responsible for elastic fiber cross-linking, lack of fibulin-4 may result in abnormal fibrillogenesis and impaired lysyl oxidase activation in the heart. Understanding the role of fibulin-4 in cardiac performance is crucial as it could provide insights into potential mechanisms underlying cardiovascular diseases like Marfan syndrome and cutis laxa syndrome, aiding in the development of targeted therapies or interventions to manage or prevent related cardiac pathologies.

In the present study, we investigated the consequences of reduced fibulin-4 expression in the mouse heart. Considering that fibulin-4[R/R] mice exhibit both cardiac and aortic characteristics akin to human pathology, encompassing aortic valve regurgitation and consequent cardiac volume overload[16], our investigation additionally focused on evaluating the cardiac vulnerability to pathology independent of aortic influence. To achieve this, we subjected fibulin-4[+/R] mice, which typically do not manifest a cardiovascular phenotype under normal circumstances, to cardiac pressure overload induced by Transverse Aortic Constriction (TAC). We furthermore conducted tissue and single-cell contractility measurements using cardiac microtissues derived from neonatal wildtype (fibulin-4[+/+]), fibulin-4[+/R], and fibulin-4[R/R] mouse hearts, alongside single adult mouse cardiomyocytes isolated from these mice. Additionally, we examined the expression of fibulin-4 within induced pluripotent stem cell-derived (iPSC) cardiomyocytes, where gene silencing experiments were performed to elucidate the potential role of fibulin-4 in maintaining cardiomyocyte homeostasis.

## Results

### A 4-fold reduced fibulin-4 expression causes cardiac remodeling and dysfunction

We monitored cardiac function in fibulin-4[R/R] mice, a validated model for aortic aneurysm formation[15]. As previously described, these homozygous fibulin-4[R/R] mice have a 75% reduction in fibulin-4 expression, which results in aneurysm formation in the aortic arch within weeks after birth and early death at about 14 weeks of age[15]. Similar to our previous observations, this fibulin-4 deficiency produced marked cardiac hypertrophy and dilation in fibulin-4[R/R] mice (Fig. 1a–f and Table S1). In addition to aortic defects, echocardiographic testing at 14 weeks showed a significant increase in left ventricular end-diastolic diameter (LVEDD) in fibulin-4[R/R] compared to fibulin-4[+/+] mice (Fig. 1c). This was associated with marked cardiac dysfunction evidenced by reductions in ejection fraction (EF) (Fig. 1d), the LV pressure-derived indices of cardiac contractility, maximum rate of rise of LV pressure (LV dP/dt$_{max}$) (Table S1) and the less afterload-sensitive rate of LV pressure increase at an LV pressure of 40 mmHg (LV dP/dt$_{P40}$) (Fig. S1a). Likewise, fibulin-4[R/R] mice showed marked cardiac diastolic dysfunction evident from a decrease in the maximum rate of fall of LV pressure (LV dP/dt$_{min}$), and the trend (p = 0,07) towards an increase (p = 0.07) in the time constant of LV pressure decay tau (Fig. S1b and Table S1). These LV systolic and diastolic abnormalities were associated with LV backward failure reflected in marked elevations in LV end diastolic pressure (LVEDP) and pulmonary congestion (Fig. S1c and Fig. S1d). Altogether, these data are consistent with our previous observations that fibulin-4-deficient mice also display a cardiac phenotype[15,16].

Histological H&E-stained sections showed that fibulin-4[R/R] mouse hearts were significantly enlarged (Fig. 1e). Accordingly, cardiomyocyte size was increased in fibulin-4[R/R] mice (Fig. 1f). LV collagen showed only minor indications of fibrosis, with 1 mouse showing exceptionally high levels (Fig. 1g). Left ventricle weight (LVW) and right ventricle weight (RVW)

were more than twice the LVW and RVW in fibulin-4[+/+] mice (LVW: 210 vs 95 mg, p < 0.0001, RVW: 44 vs 21 mg, p < 0.001), and this remained significant after correction for tibia length (Fig. 1h, i). The mRNA expression of the hypertrophy markers atrial natriuretic peptide (ANP), brain natriuretic peptide (BNP), and alpha-skeletal actin (αSKA) was increased, along with reduced expression of sarcoplasmic/endoplasmic reticulum Ca$^{2+}$-ATPase (SERCA2a) (Fig. 1j–m), consistent with cardiac hypertrophy and dysfunction. Together, the data show pathological remodeling in the fibulin-4[R/R] mouse heart.

To investigate the direct impact of fibulin-4 on cardiac cell function, we analyzed cardiac tissue in an isolated in vitro setting. For this purpose, we cultivated in vitro cardiac tissue derived from neonatal fibulin-4[+/+] and fibulin-4 deficient mouse hearts using a specialized micro-system (Fig. 2a). This system allowed further evaluation of the role of fibulin-4 on cardiac structural organization and cardiomyocyte contractile behavior in an in vitro cardiac model system mimicking 3D cardiac tissue[23,24]. After producing these microtissues from cardiac cells of neonatal fibulin-4[+/+], fibulin-4[+/R] and fibulin-4[R/R] hearts, we found that, based on the vimentin staining, fibulin-4[+/R] and fibulin-4[R/R] microtissues contained significantly more cardiac fibroblasts after 4 days (35 ± 1% and 33 ± 1%) compared to microtissues generated from cardiac cells of fibulin-4[+/+] hearts (26 ± 2%) (Fig. 2b, f). Furthermore, fibulin-4[R/R] and fibulin-4[+/R] cardiomyocyte sarcomere length was not different compared to fibulin-4[+/+] microtissues (Fig. 2c).

To investigate the effect of fibulin-4 deficiency on cardiac function, beating frequency and dynamic contraction force were measured for microtissues of all genotypes at day 2. This functional analysis revealed that after 2 days the beating frequency in both fibulin-4[+/R] and fibulin-4[R/R] microtissues was significantly decreased (Fig. 2d). Additionally, while dynamic contraction force was unaffected in fibulin-4[+/R] microtissues, the dynamic contraction force of fibulin-4[R/R] microtissues was significantly impaired compared to fibulin-4[+/+] microtissues (Fig. 2e). After 4 days, both beating frequency and dynamic contraction force of the fibulin-4[+/R] microtissues showed a trend towards decreasing to levels comparable with the fibulin-4[R/R] microtissues (Fig. 2d, e). To further assess the effect of reduced fibulin-4 expression on cardiomyocyte function we measured isometric force of single membrane-permeabilized adult cardiomyocytes isolated from the fibulin-4[+/+], fibulin-4[+/R] and fibulin-4[R/R] mice (Fig. 2g), at various calcium concentrations (Fig. 2h). Similar to results from the microtissues, fibulin-4[R/R] but not fibulin-4[+/R] decreased maximal cardiomyocyte force generation (F$_{max}$) by 20% (Fig. 2i). No changes in passive force (F$_{pas}$) (all ~4 kN/m$^2$) or calcium-sensitivity (pCa$_{50}$) (all ~10$^{-5,6}$ M) were observed. Together, these results demonstrate that in addition to proper elastogenesis, fibulin-4 regulates cardiac muscle function and cardiomyocyte contractility.

### Reduced fibulin-4 expression increases cardiac susceptibility to pressure-overload

We subsequently evaluated fibulin-4 deficiency-mediated cardiac defects in vivo, independent of aortic abnormalities, in 14-week-old heterozygous fibulin-4[+/R] mice. Fibulin-4[+/R] animals have only a 2-fold reduction in fibulin-4 expression and demonstrate no cardiac or aortic valve dysfunction under normal conditions[15]. To challenge the hearts of fibulin-4[+/R] mice we induced cardiac pressure-overload through severe transverse aortic constriction (sTAC) or mild transverse aortic constriction (mTAC). After 4 weeks of sTAC, reduced fibulin-4 expression markedly aggravated sTAC-induced mortality from 20% in fibulin-4[+/+] mice to 80% in fibulin-4[+/R] littermates (Fig. 3a). Because of the very high mortality rate in sTAC fibulin-4[+/R] mice, we did not try to increase the number of surviving mice to a level that would allow evaluation of cardiac function or molecular analysis. Cardiac dimension and function of sTAC mice surviving 4 weeks after surgery are shown in Fig. S2.

Four weeks of mTAC resulted in 27% mortality in fibulin-4[+/R] mice without affecting survival in fibulin-4[+/+] animals (Fig. 3a). Accordingly, a 2-fold reduction in fibulin-4 expression by itself did not affect heart size

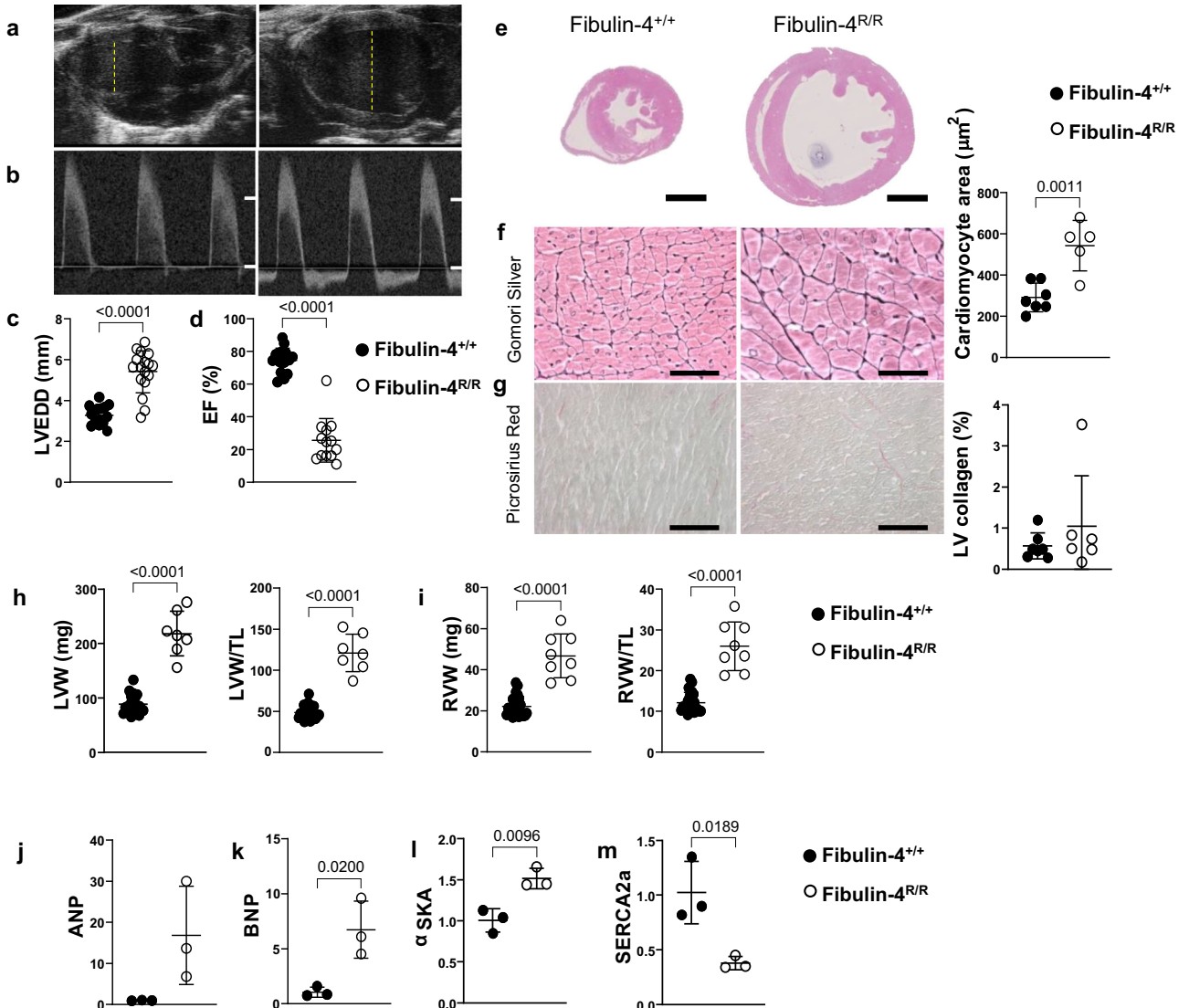

**Fig. 1 | Comparison between fibulin-4$^{R/R}$ and fibulin-4$^{+/+}$ mouse hearts.**
**a** Representative B-mode images from the left ventricle of 18-week-old fibulin-4$^{+/+}$ (n = 16) and fibulin-4$^{R/R}$ male mice, 4 weeks post-TAC (n = 6). Yellow dotted line represents left ventricle diameter. **b** Aortic flow patterns (white lines represent 0 and 1000 mm/s). **c** Measured left ventricular end diastolic diameter (LVEDD) and (**d**) calculated ejection fraction (EF). **e** Representative cross section of mouse hearts. Black scale bars represent 2 mm. **f, g** Gomori Silver and Picrosirius Red stainings were used to determine cardiomyocyte size and collagen deposition in the left

ventricle, respectively. Black scale bars represent 200 μm. **h, i** Weights of left and right mouse ventricles, without and with normalization to tibia length (TL). **j–m** Relative mRNA expression of atrial natriuretic peptide (ANP), brain natriuretic peptide (BNP), α-skeletal muscle actin (α-SKA), sarcoplasmatic reticulum Ca$^{2+}$-ATPase (SERCA2a), normalized to fibulin-4$^{+/+}$ hearts, in fibulin-4$^{+/+}$ (n = 3–7) and fibulin-4$^{R/R}$ 18-week-old male mice (n = 3–4). Lines indicate mean ± SEM. *p < 0.05 vs corresponding genotype. A statistical *t* test was performed.

(Fig. 3b, h, i), LVEDD (Fig. 3j) or aortic valve function (Fig. 3d) in sham mice, whereas cardiac hypertrophy and remodeling produced by mTAC was aggravated in surviving fibulin-4$^{+/R}$ mice (Fig. 3b, c, e, f) without inducing aortic valve regurgitation in either genotype (Fig. 3d). This was reflected in a trend towards increased relative LVW (p = 0.14) and a significant increase in LVEDD (Fig. 3h and Fig. 3j). Additionally, mTAC-induced cardiac dysfunction (decrease in EF) was markedly aggravated in mice with reduced fibulin-4 expression (Fig. 3k), while mTAC only affected LV dP/dt$_{P40}$ and LV dP/dt$_{max}$ in fibulin-4$^{+/R}$ animals but not in fibulin-4$^{+/+}$ littermates (Fig. 3l, m). Similarly, only in fibulin-4$^{+/R}$ mice, mTAC produced diastolic dysfunction as evidenced by an increase in LVEDP and a decrease in LV dP/dt$_{min}$ (Fig. 3l, n), as well as an increased relative RVW indicative of left-sided backward heart failure (Fig. 3i). Aorta diameter and distensibility were not affected by mTAC in fibulin-4$^{+/R}$ mice (Fig. 3f, g). Thus, fibulin-4$^{+/R}$ mice do not produce cardiovascular adaptations under normal circumstances but show markedly aggravated mortality and LV remodeling and

dysfunction following cardiac pressure-overload, in the absence of any perturbations in aortic valve function.

## Impact of reduced fibulin-4 expression on extracellular matrix and cardiac remodeling
Our subsequent analyses aimed to assess the impact of reduced fibulin-4 expression on the composition of the extracellular matrix. Gomori silver staining showed an increased cardiomyocyte size after mTAC (Fig. 4a, c). Increased collagen content after mTAC was observed in both fibulin-4$^{+/+}$ and fibulin-4$^{+/R}$ hearts as determined by picrosirius red staining (Fig. 4b). To accurately measure the degree of interstitial fibrosis in the myocardium in response to mTAC, the percentages of fibrotic tissue was determined in the captured images using a quantitative image analysis system. The interstitial collagen volume fraction was significantly increased in post-TAC fibulin-4$^{+/R}$ compared to post-TAC fibulin-4$^{+/+}$ mouse hearts (Fig. 4d). Our findings indicate increased fibrosis in fibulin-4$^{+/R}$ hearts after mTAC.

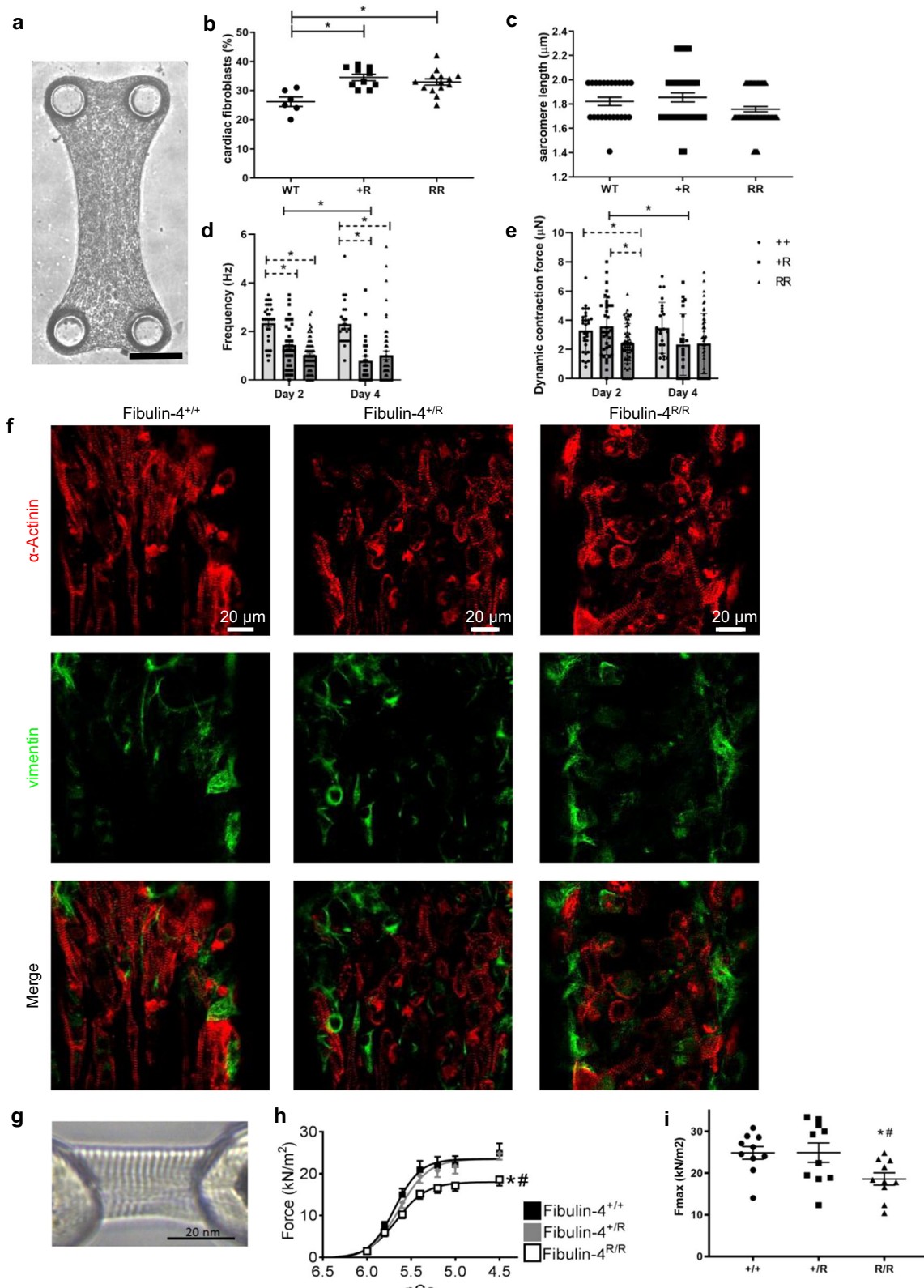

Subsequently, we assessed markers associated with hypertrophy and fibrosis. Initially, at baseline, the mRNA expression levels of ANP, BNP, and αSKA did not exhibit an elevation in fibulin-4[+/R] hearts when compared to fibulin-4[+/+] (Fig. 4e–g). However, following mTAC induction, there was a notable and significant increase in ANP, BNP, and αSKA mRNA expression specifically in the fibulin-4[+/R] mice, contrasting with the unaffected levels observed in fibulin-4[+/+] mice (Fig. 4e–g). While connective tissue growth factor (CTGF) mRNA expression remained unchanged after mTAC in fibulin-4[+/+] mice, a substantial increase was detected in the fibulin-4[+/R] mice post-mTAC compared to sham-operated mice, indicating the initiation of fibrosis (Fig. 4h). Notably, no statistically significant alterations were observed in SERCA2a expression (Fig. 4i). To unravel the role of the

**Fig. 2 | Cardiac tissue models of isolated fibulin-4$^{+/+}$, fibulin-4$^{+/R}$ and fibulin-4$^{R/R}$ mouse heart. a** Example of a microtissue. Black scale bar represents 50 μm. **b, c** Percentage of cardiac fibroblasts (fibulin-4$^{+/+}$ n = 6, fibulin-4$^{+/R}$ n = 10, fibulin-4$^{R/R}$ n = 14) and sarcomere length (fibulin-4$^{+/+}$ n = 24 sarcomeres from 6 images, fibulin-4$^{+/R}$ n = 40 sarcomeres from 10 images, fibulin-4$^{R/R}$ n = 56 sarcomeres from 14 images) in 4-day old cardiac microtissues. Bars represent 20 μm, *p < 0.05 vs corresponding fibulin-4$^{+/+}$, #p < 0.05 vs corresponding fibulin-4$^{+/R}$, data is presented as mean ± SEM. Statistical analyses were performed using a non-parametric Kruskal-Wallis test. **d, e** Beating frequency and dynamic contraction force for 4-day old cardiac microtissues (fibulin-4$^{+/+}$ n = 29, fibulin-4$^{+/R}$ n = 35, Fibulin-4$^{R/R}$ n = 63).

**f** Cell distribution and sarcomere organization in 4-day-old cardiac microtissues, stained for α-actinin (red) and vimentin (green). Note that most fibroblasts were found along the tissue edges. White scale bar represents 20 μm. **g** Example of a single adult cardiomyocyte mounted in the setup, isolated from the left ventricle (LV). Black scale bar represents 20 nm. **h, i** Functional analyses of 3 LVs per group; 2–4 cells per LV. Black scale bar represents 20 nm. Maximal force, Fmax; *p < 0.05 vs corresponding fibulin-4$^{+/+}$, #p < 0.05 vs corresponding fibulin-4$^{+/R}$. Data is presented as mean ± SEM, a non-parametric Kruskal–Wallis test was performed for the microtissue analysis, isometric force differences were analyzed using a statistical t test.

angiotensin II (AngII) system in the effects of fibulin-4 on cardiac pathology, we additionally determined plasma renin concentration (PRC). Due to large variations in the measurements, the log means of each group was back-transformed (antilog) and, given the properties of logarithms, are an estimate of the PRC (Fig. 4j). Overall, PRC was elevated by TAC (p = 0.02, sham vs mTAC) but no statistical differences were observed between fibulin-4$^{+/+}$ and fibulin-4$^{+/R}$ mice (p = 0.56, Fig. 4j). The elevation in collagen and hypertrophy markers signifies an escalated presence of fibrosis and hypertrophy following mTAC, a condition exacerbated by the reduction in fibulin-4 expression.

### Exploration of cardiac remodeling processes and implications of fibulin-4 expression

Subsequently, we investigated the underlying processes contributing to cardiac remodeling by examining genes implicated in extracellular matrix (ECM) remodeling, particularly focusing on the TGF-β pathway signaling and mechanosensing mechanisms. Notably, sham-operated fibulin-4$^{+/R}$ mice exhibited a tendency toward reduced fibulin-4 expression compared to fibulin-4$^{+/+}$ sham mice (Fig. 5a, p = 0.652). Strikingly, following mTAC, both fibulin-4$^{+/+}$ and fibulin-4$^{+/R}$ mice displayed increased fibulin-4 expression at both mRNA and protein levels (Fig. 5a, d, e). Additionally, alongside elevated fibulin-4 levels, elastin (ELN) mRNA expression also showed an upregulation, while plasminogen activator inhibitor-1 (PAI1) mRNA levels, a TGFβ-activated gene, remained similar to fibulin-4$^{+/+}$ (Fig. 5b, c). Notably, under normal conditions, fibulin-4 expression in the heart is nearly negligible, resulting in minimal detectable protein levels in the sham condition for both mouse groups, rendering any discernible difference undetectable (Fig. 5d, e, p = 0.996). However, mTAC induced a comparable increase in fibulin-4 expression in both mouse groups (Fig. 5e).

Activation of Smad2/Smad3 has established involvement in cardiac fibrosis[25], while extracellular signal-regulated kinase (ERK) signaling contributes to the hypertrophic response to wall stress[26]. Therefore, we assessed the phosphorylation status of Smad2 and ERK in control and TAC treated fibulin-4$^{+/+}$ animals and fibulin-4$^{+/R}$ mice. Our findings revealed no alterations in the phosphorylation levels of either Smad2 or ERK, suggesting that neither mTAC nor reduced fibulin-4 expression influenced TGF-β signaling under these conditions in cardiac tissue (Fig. 5f–i).

Previous studies have indicated reduced FAK activation in mice with dilated cardiomyopathy[4]. In our investigation, FAK activation displayed a reduction post-mTAC regardless of fibulin-4 expression (Fig. 5j, k). This reduction was notably significant in fibulin-4$^{+/+}$ mice, while a similar trend was observed in fibulin-4$^{+/R}$ mice (Fig. 5k, mTAC fibulin-4$^{+/R}$ vs Sham fibulin-4$^{+/R}$; p = 0.102). Notably, reduced fibulin-4 expression did not directly influence FAK activation (Fig. 5k, sham fibulin-4$^{+/R}$ vs sham fibulin-4$^{+/+}$; p = 0.836), nor did it result in significant differences in total FAK expression across conditions. Newborn fibulin-4$^{+/R}$ mice exhibit an ~25% decreased baseline mRNA expression of fibulin-4 in the aorta[15]. Interestingly, following mTAC of adult (18-weeks old) mice, we observed a notable increase in fibulin-4 expression in heart tissue (Fig. 5e). This implies that the decreased fibulin-4 expression in fibulin-4$^{+/R}$ mice can still adequately sustain cardiac function, albeit with a slight reduction in survival rates after mTAC (Fig. 3a). Additionally, these fibulin-4$^{+/R}$ mice retain the capacity to increase fibulin-4 expression levels following the mTAC procedure.

However, when subjected to sTAC, fibulin-4$^{+/R}$ mice exhibited a significantly lower survival rate compared to both fibulin-4$^{+/+}$ mice subjected to sTAC and fibulin-4$^{+/R}$ mice subjected to mTAC (Fig. 3a). Consequently, this prompted a more extensive analysis of mice undergoing sTAC to further delineate the impact of this procedure on cardiac function in the context of reduced fibulin-4 expression.

### Fibrotic response and activation of cardiac fibulin-4 in response to cardiac pressure overload

To investigate the variations in histopathological and morphologic alterations in the sTAC treated animals, H&E and Masson's trichrome staining were applied to evaluate cardiomyocyte viability and fibrosis in the different groups. Substantial cellular infiltrate in the interstitium and disorganized cardiac muscle fibers were observed in the sTAC-treated mice, compared with sham mice (Fig. 6a). Next to increased interstitial fibrosis in response to the TAC procedure, we observed more extensive areas with replacement fibrosis following sTAC in both genotypes (Fig. 6b).

To establish if the exaggerated dilated cardiomyopathy triggered by TAC in fibulin-4$^{+/R}$ mice with reduced gene dosage originates primarily from abnormalities within the extracellular matrix, we investigated the presence of fibulin-4 protein and elastin expression in the myocardium of fibulin-4$^{+/+}$ and fibulin-4$^{+/R}$ animals. Where fibulin-4 was not detectable in the myocardium of untreated adult mice, in fibulin-4$^{+/+}$ animals subjected to sTAC, fibulin-4 expression was activated across the entire myocardium (Fig. 6c). This upregulation of fibulin-4 expression in response to sTAC was not observed in fibulin-4$^{+/R}$ myocardium (Fig. 6e). Elastin expression appeared to be increased after sTAC in both genotypes (Fig. 6d and Fig. 6f). However, elastin in fibulin-4$^{+/+}$ mice showed a more mature appearance surrounding the cardiomyocytes whereas in fibulin-4$^{+/R}$ animals elastin appeared in a more immature patchy pattern.

To obtain insight in the potential origin of the deposition of fibulin-4 surrounding the cardiomyocytes, we analyzed the single-cell data published by Froese et al. in TAC mice[27]. RNA-sequencing was performed on cardiomyocytes, fibroblasts, and endothelial cells[27]. We found that, in their data, expression of fibulin-4 was upregulated after 1 week of TAC in all three cell types compared to sham (Fig. S3). Unexpectedly, while fibroblasts and endothelial cells showed a transient early upregulation of fibulin-4 expression, which was diminished after 8 weeks of TAC, cardiomyocytes maintain a high level of expression compared to 1-week sham mice.

### Knockdown of fibulin-4 affects cardiac gene expression and cellular dimensions in human iPSC-derived cardiomyocytes

As we observed elevation in fibulin-4 expression encompassing cardiomyocytes within the myocardium subsequent to TAC treatment in vivo, we aimed to determine whether fibulin-4 expression was associated with fibroblasts or cardiomyocytes. We next examined iPSC-derived cardiomyocytes, to understand these autonomous effects. Within the cardiomyocyte cultures derived from iPSCs, fibulin-4 demonstrated intrinsic expression, predominantly localized within the extracellular matrix (ECM) (Fig. 7a). Notably, fibulin-4 expression exhibited a higher presence within the ECM surrounding cells that did not exhibit positive α-actinin staining (non-cardiomyocytes), while a modest level of expression was also detected within the cardiomyocytes themselves (Fig. 7a). This led us to postulate that

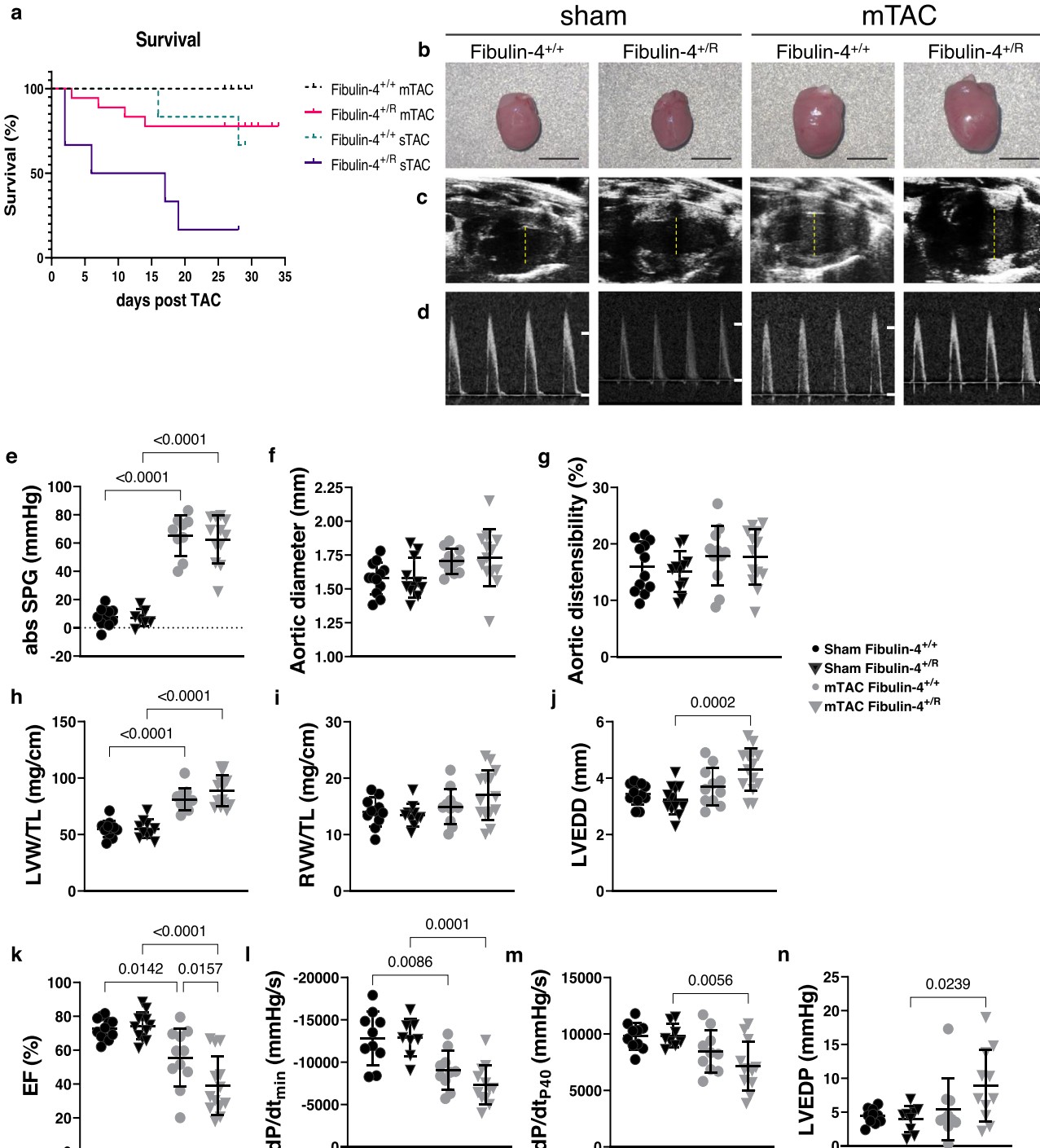

**Fig. 3 | Performance of 18-week old mouse hearts 4 weeks post-TAC. a** Survival of fibulin-4$^{+/+}$ and fibulin-4$^{+/R}$ male mice with mild TAC (mTAC) or severe TAC (sTAC). Tick marks represent censored observations. **b** Representative whole hearts of mTAC mice. Black scale bars represent 4 mm. **c** LV long axis B-mode images of 18-week-old male mouse hearts 4 weeks post-TAC (Yellow dotted line represents left ventricle diameter). **d** Aortic flow patterns (white lines represent 0 and 1000 mm/s). **e–n** Hemodynamic measurements of mTAC mice. Lines represent mean ± SEM. Dots and triangles resemble individual mice. Kaplan–Meier statistical analyzed was performed on the survival curves, cardiac dimension and function data were tested using a two-way ANOVA. systolic pressure gradient (SPG), left ventricle weight (LVW), right ventricle weight (RVW), tibia length (TL), left ventricular end-diastolic diameter (LVEDD), ejection fraction (EF), maximum rate of fall in LV pressure (dP/dt$_{min}$), rate of rise in LV pressure at LV pressure of 40 mmHg (dP/dt$_{p40}$), maximal rise of pressure (dP/dt$_{max}$), left ventricle end diastolic pressure (LVEDP).

cardiomyocytes might possess the capability to generate ECM to a certain extent, raising the possibility that interference with fibulin-4 could impact cardiomyocyte equilibrium. Using lentivirus-mediated transfer of two distinct shRNAmirs targeting human fibulin-4, we successfully reduced fibulin-4 mRNA levels by 72% and 88%, respectively (Fig. 7c). This reduction in fibulin-4 expression corresponded with an increase in cardiomyocyte surface area by ~40% (Fig. 7b, d), akin to the observed cardiomyocyte hypertrophy noted in fibulin-4$^{R/R}$ hearts (Fig. 1f).

To investigate whether the reduction of fibulin-4 affected cardiomyocyte behavior in vitro, we next determined RNA expression levels of

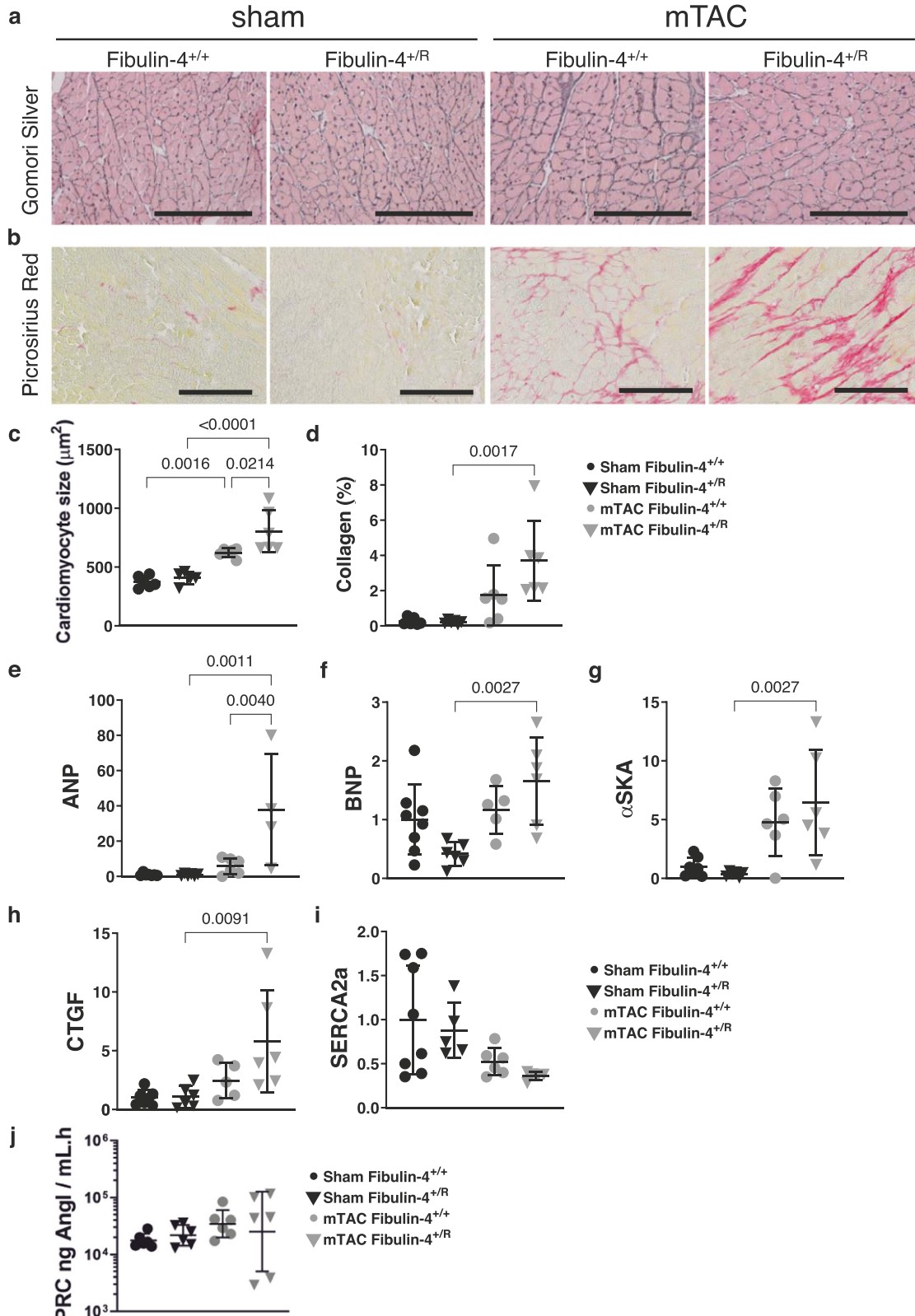

**Fig. 4 | Hypertrophic remodeling of 18-week-old male mice hearts 4 weeks post-mTAC. a, b** Histological stainings of paraffin embedded LV tissue (N = 6). **c, d** Cardiomyocyte size and collagen deposition were quantified using the Gomori Silver and Picrosirius Red staining, respectively. Black scale bars represent 200 μm. **e–i** mRNA expression of atrial natriuretic peptide (ANP), α-skeletal muscle actin (αSKA), sarcoplasmic reticulum Ca²⁺-ATPase (SERCA2a) and connective tissue growth factor C (CTGF) in sham (fibulin-4$^{+/+}$ n = 7–8, fibulin-4$^{+/R}$ n = 5–6) and mTAC mice (fibulin-4$^{+/+}$ n = 5–6, fibulin-4$^{+/R}$ n = 4–6), (all male). Lines represent mean ± SEM. **j** Plasma renin concentrations (PRC) in sham and mTAC fibulin-4$^{+/+}$ and fibulin-4$^{+/R}$ mice, (n = 6, all male) in all groups. PRC data represents geometric mean *÷ geometric SD. Two-way ANOVA testing was performed.

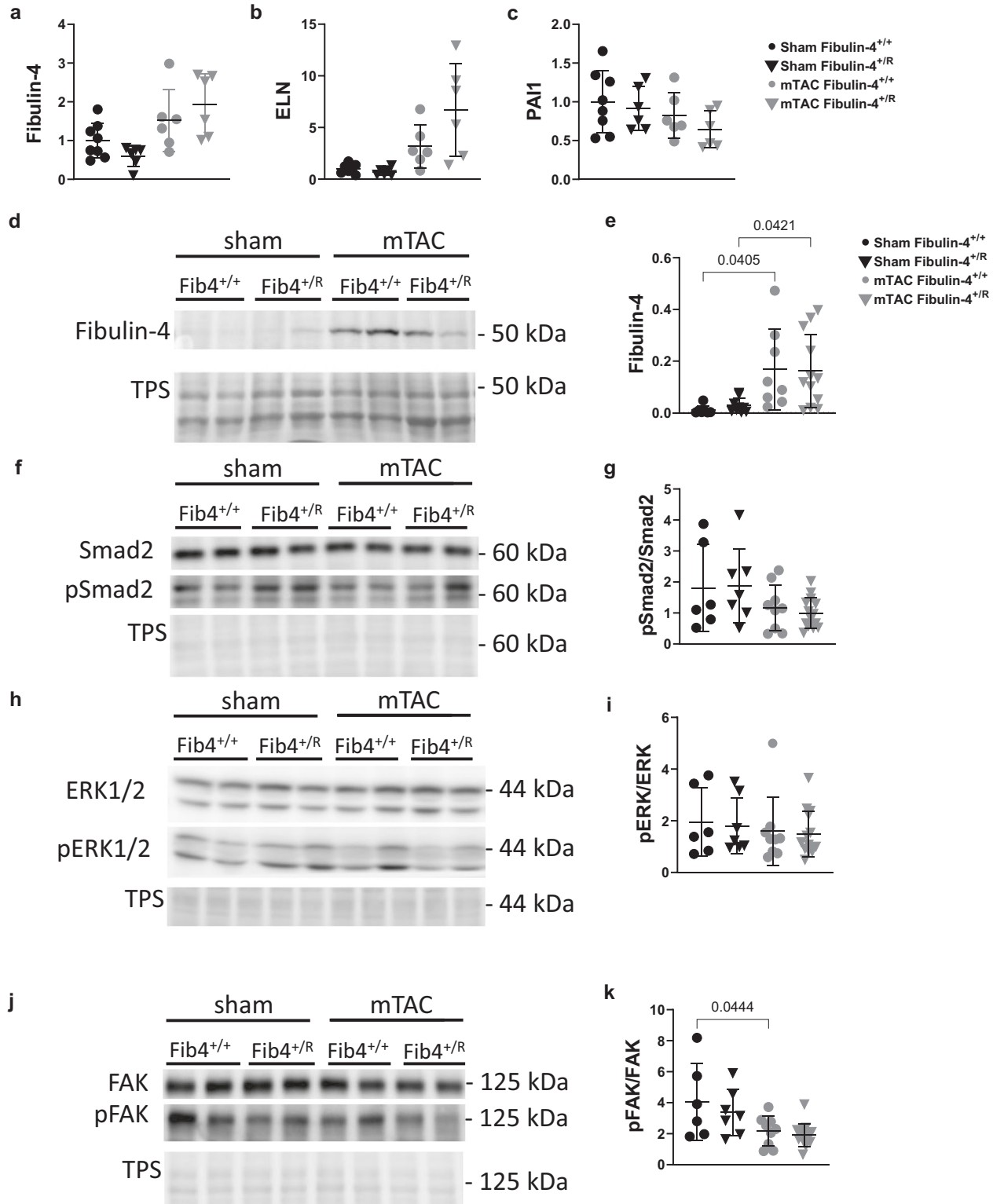

**Fig. 5 | Effect of mTAC on fibulin-4, elastin (ELN), PAI1, (phospho)Smad2, (phospho)FAK, and (phospho)ERK1/2 expression in fibulin-4$^{+/+}$ and fibulin-4$^{+/R}$ 18-week-old male mouse heart samples, 4 weeks post-TAC. a–c** mRNA expression of fibulin-4, ELN, and PAI1. **d–k** Representative blots and quantified protein expression of (**d**, **e**) fibulin-4, (**f**, **g**) (phospho) Smad2, (**h**, **i**) (phospho) focal adhesion kinase (FAK), (**j**, **k**) (phospho) extracellular signal-regulated kinase (ERK). Lines represent mean ± SEM. A two-way ANOVA testing was performed. TPS total protein stain.

cardiac hypertrophy-associated genes. We observed an increase in mRNA expression of TGF-β activated genes PAI-1 and CTGF, associated with cardiomyocyte hypertrophy (Fig. 7e, f), as well as an increase of ANP mRNA expression, a marker for cardiac stress (Fig. 7g). Similarly, the Myh7/Myh6 ratio, indicative of cardiac dysfunction, showed a small, albeit insignificant

increase subsequent to fibulin-4 knockdown (Fig. 7h, shRNA Fibulin-4#1: p = 0.21, shRNA Fibulin-4#2: p = 0.11). Taken together, these findings support the concept that a decrease in fibulin-4 levels produces a gene-expression pattern in iPSC-derived cardiomyocytes that is associated with cardiomyocyte hypertrophy and dysfunction.

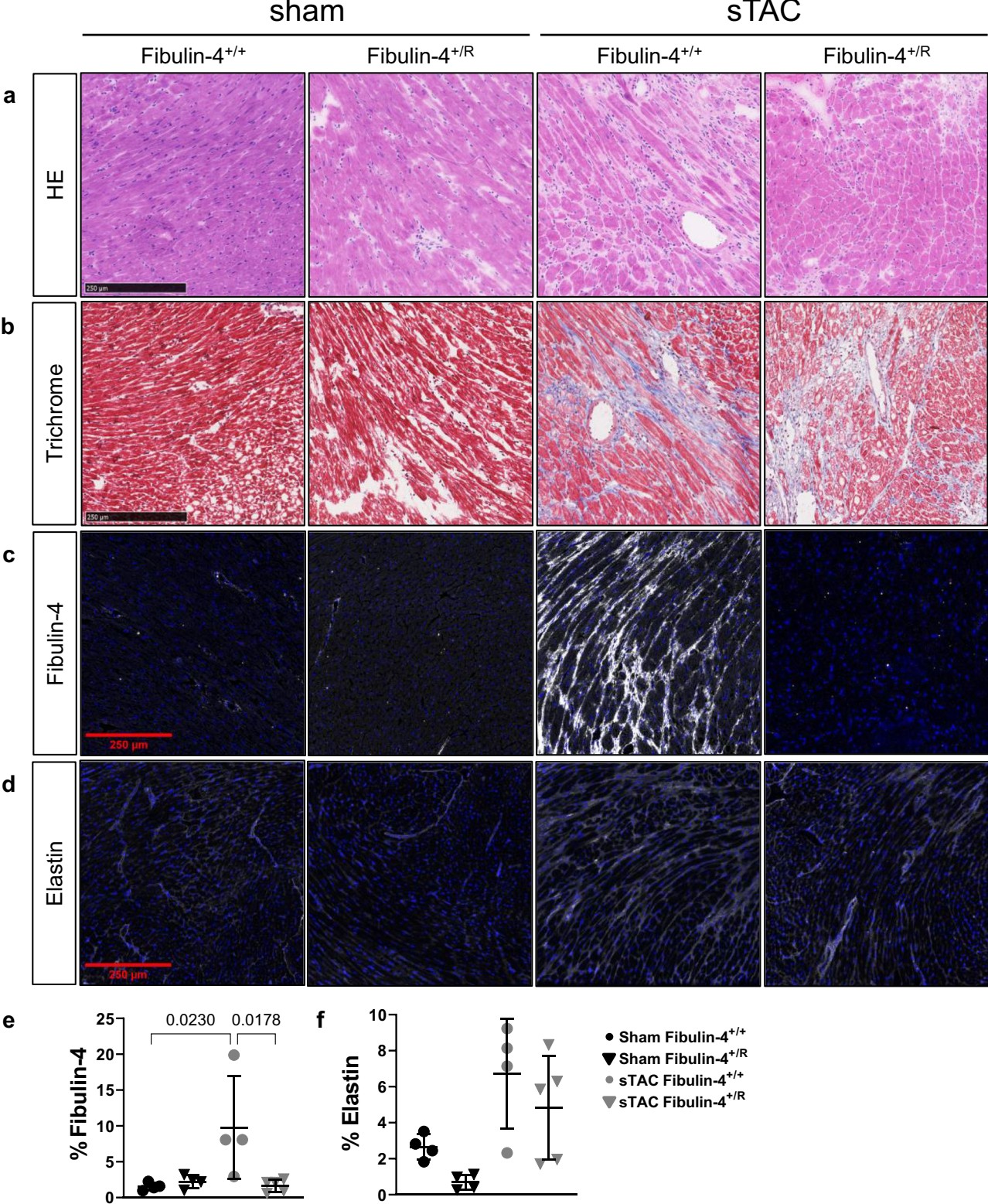

**Fig. 6 | Heart tissue of fibulin-4$^{+/+}$ and fibulin-4$^{+/R}$ 18-week-old male mice 4 weeks post-sTAC. a** Representative histological staining of heart tissue, **b** staining for fibrosis, (**c, d**) as well as immunostainings of elastin and fibulin-4. **e, f** Quantification of fluorescent signal of fibulin-4 and elastin. Black scale bars represent 250 μm. Lines represent mean ± SEM. A two-way ANOVA testing was performed.

## Discussion

In the present study, we identified fibulin-4 as a pivotal factor in cardiac dysfunction irrespective of extrinsic vascular or valve abnormalities. Fibulin-4$^{R/R}$ mice with severely reduced fibulin-4 expression show increased LV diameter and mass, severe cardiac dysfunction, and altered expression of the cardiac hypertrophy failure-associated genes ANP, BNP, α-SKA, and SERCA2a. Even though cardiac pathology in fibulin-4$^{R/R}$ mice is likely aggravated by the concurrent aortic valve disease, the development of cardiac remodeling and dysfunction was more pronounced in fibulin-4$^{R/R}$ mice than in other genetic or mechanically-induced models of aortic

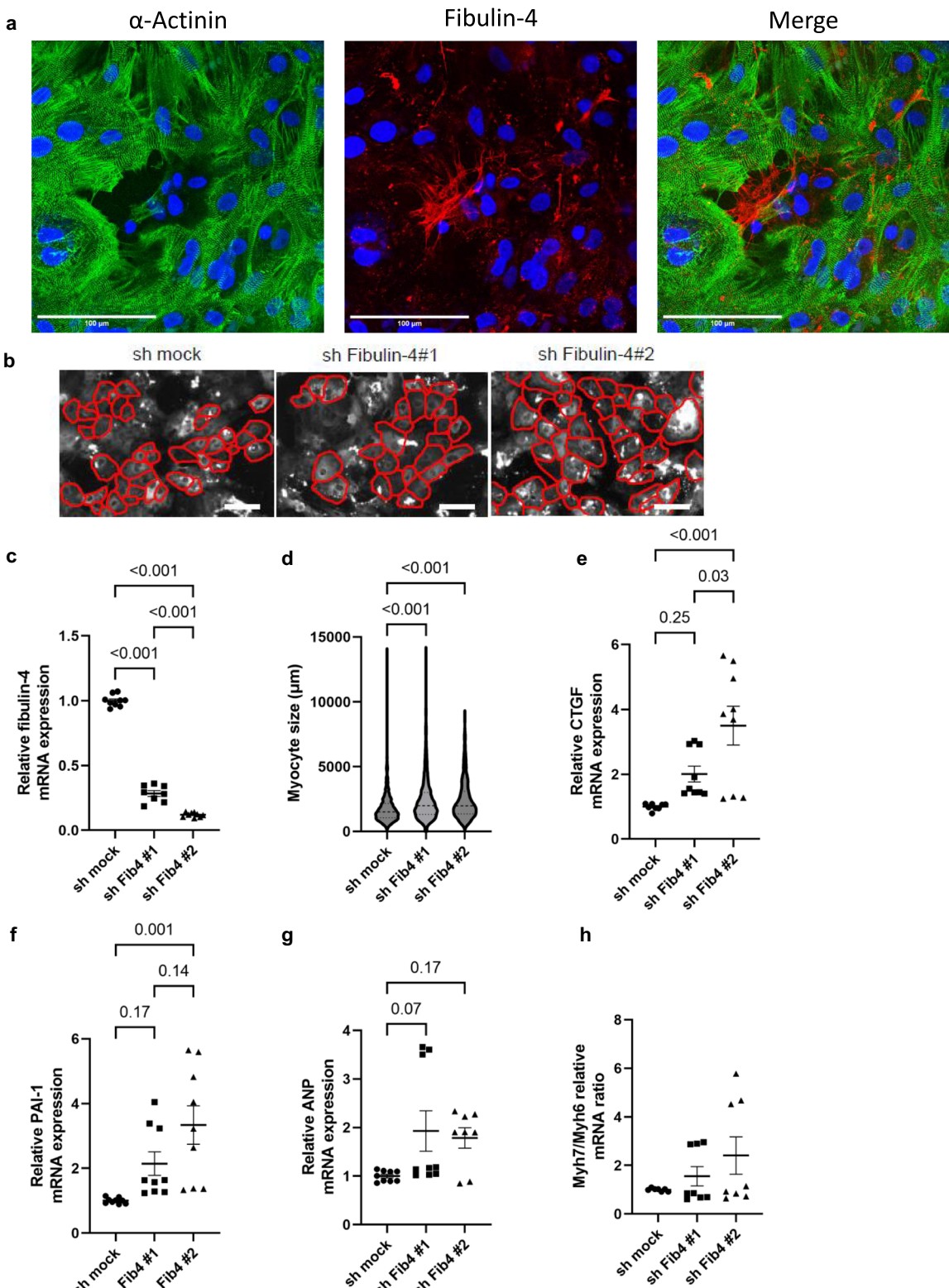

**Fig. 7 | RNA interference of fibulin-4 in human iPSC-cardiomyocytes. a** Fibulin-4 expression in the ECM surrounding human iPSC-derived cardiomyocytes and non-cardiomyocytes; α-actinin (green), fibulin-4 (red), nucleus (blue). **b** Example images with segmentation of human iPSC-cardiomyocytes. In a total of 40–46 fields of view per group, 1513 shRNA mock, 1081 shRNA Fibulin-4#1, and 830 shRNA Fibulin-4#2 transfected cells were analyzed. **c** Relative fibulin-4 mRNA expression after transfection. **d** Myocyte sizes after transfection. **e–h** Relative mRNA expression of genes related to hypertrophy. White scale bars represent 100 μm. Lines represent mean ± SEM. One-way ANOVA testing was performed.

regurgitation[17–20]. This suggests that at least part of the cardiac disease in fibulin-4[R/R] mice is related to a primary effect of fibulin-4 on cardiac dysfunction. In addition, longitudinal analysis of aneurysmal growth in fibulin-4[R/R] mice showed that aortic and left ventricular growth occur simultaneously. Interestingly, losartan reduced left ventricle growth specifically, while having a limited effect on aneurysmal growth[28]. This indicates that aortic and left ventricle pathology occur, at least in part, independently. In light of this discovery, it would be interesting to see if losartan treatment could rescue the aggravated TAC-phenotype of the fibulin-4[+/R] mice, where aortic insufficiency does not play a role, but this is beyond the scope of the current research.

The concept of fibulin-4 deficiency having a primary effect on cardiac dysfunction was further supported by the observation that fibulin-4[+/R] mice showed an increased susceptibility to cardiac pathology upon cardiac pressure overload while maintaining normal aortic valve function. Thus, similar to observations from our previous study[15,16], the size of the ascending aorta as well as the anatomy and function of the aortic valves were not affected in fibulin-4[+/R] mice. Similarly, we found no significant differences in aortic distensibility or diameter between fibulin-4[+/+] and fibulin-4[+/R] mice with or without TAC. In these animals, reduced fibulin-4 expression increased mortality and aggravated TAC-induced cardiac hypertrophy and dysfunction, as well as expression of heart failure-associated genes. In line with our previous findings in fibulin-4[R/R] mice, in fibulin-4[+/R] mice we observed a strong trend towards activation of ERK1/2, which, as a member of the mitogen-activated protein kinase family, is involved in cardiac remodeling and dysfunction[29,30]. However, the impact of altered fibulin-4 expression on ERK1/2 was annulled by TAC. An aorta independent direct influence of fibulin-4 on the heart was also demonstrated in microtissues derived from neonatal cardiac cells of fibulin-4[+/+], fibulin-4[+/R], and fibulin-4[R/R] mice, where reduced fibulin-4 expression promoted fibroblast proliferation. Additionally, cardiac microtissues with reduced fibulin-4 expression demonstrated decreased beating frequency and contractile force, further proving that reduced fibulin-4 expression induces primary myocardial dysfunction. These results were further corroborated by measurements in single cardiomyocytes from adult mice with reduced fibulin-4 expression, that similarly showed an impaired force generating capacity.

The observed differences in cardiac responses between mTAC and sTAC conditions in fibulin-4[+/R] mice compared to WT mice suggest that the initial ECM deposited during development and maturation may be inherently weaker in these mice, which could impair their ability to cope with chronic overload. Fibulin-4 is essential for lysyl oxidase activation, which is crucial for effective collagen cross-linking[6,31]. Therefore, a weaker ECM in fibulin-4[+/R] mice, due to impaired cross-linking, likely contributes to their sensitivity to stress-induced conditions. This ECM weakness may also explain the poorer outcomes observed in these mice under pressure overload scenarios. Although we did observe a difference in elastin in sTAC mouse hearts compared to sham, we did not further examine the integrity of the deposited ECM. Further characterization of elastin assembly would be necessary to fully understand the mechanistic defects which occur during pressure overload.

The origin of fibulin-4 expression also remains to be determined. Our data shows that both myocyte and non-myocyte cells (fibroblasts, endothelial cells) express fibulin-4. We assume that myocyte-specific expression is a response to hypertrophy. Data from Froese et al. showed that fibulin-4 (*Efemp2*) expression is increased in response to TAC in mice (Fig. S3)[27]. Cardiomyocyte expression remained high, particularly compared to fibroblasts, suggesting that cardiomyocytes do contribute to remodeling in response to increased cardiac load. Other cell types, such as macrophages, may also be important regulators of fibulin-4 expression. Data available from the study by Ren et al. shows that fibulin-4 expression seems to be increased in macrophages after TAC[32]. Additionally, smooth-muscle cell-specific knockout of *Efemp2* results in poorly organized fibrils[8]. Further characterization of cell-specific expression of fibulin-4 and other ECM components during TAC is required to completely understand the individual roles of these cells during remodeling.

While previous research with fibulin-4[+/R] mice has shown a 50% reduction in fibulin-4 expression, we only observed a trend towards reduction in our mRNA experiments (Fig. 5a). Previous studies measured the levels of fibulin-4 in the kidney[16]. A difference in location of the tissue sample of which the mRNA was isolated could explain part of this difference in fibulin-4 expression. However, it seems more likely that the heterogeneity in both mRNA and protein expression between mice is caused by varying degrees of hypertrophy. Additionally, we measured mRNA expression in heart samples of adult mice. Due to the nature of the construct, which leaves regulatory sequences of fibulin-4 intact, fibulin-4 may be re-expressed in response to TAC.

In individuals with Marfan syndrome or cutis laxa, cardiac failure is generally explained as a consequence of aortic valve disease and thoracic aneurysms. Cardiomyopathy in fibrillin 1-deficient mice was demonstrated to result from a primary cardiac impairment that appeared to be caused by ECM-induced abnormal cardiomyocyte mechano-signaling[4]. In our mouse model, a reduction in FAK activation was measured in mTAC-treated mice compared to sham mice, independent of fibulin-4 expression. In fibrillin-1 deficient mice, FAK activation was reduced. This indicates that, in contrast to fibrillin-1, fibulin-4 is unlikely to be directly involved in mechano-sensing in the cardiac wall.

A functional role for fibulin-4 in addition to elastogenesis has also been shown in smooth muscle specific fibulin-4 knockout mice, which showed a failure in aortic wall differentiation, marked by reduced expression of smooth muscle-specific contractile genes, focal proliferation of smooth muscle cells, and degenerative changes in the medial wall. In our microtissues, neonatal fibulin-4[+/+] cardiomyocytes were more elongated and oriented in one direction, whereas cardiomyocytes in fibulin-4[+/R] and particularly fibulin-4[R/R] microtissues appeared more rounded. Yet, sarcomere length was similar for all groups. The elongated shape may influence sarcomere organization, which is essential for proper cardiomyocyte contraction[33]. Accordingly, we observed a dose-response effect of reduced fibulin-4 expression on the spontaneous beating frequency in the microtissues. Furthermore, the dynamic contraction force was reduced in fibulin-4[R/R] cardiac microtissues. In line with these observations, the maximal generated force was significantly lower in single adult permeabilized cardiomyocytes from fibulin-4[R/R] mice compared to fibulin-4[+/+] littermates.

A potential physiological mechanism linking fibulin-4 to cardiac dysfunction could also be found in the indirect effects of fibulin-4 on actin. Fibulin-4 deficiency increases the activity of the actin depolymerizing factor cofilin[22]. Activation of cofilin can result in cytoskeletal disassembly as well as sarcomere dysfunction through shortening thin filament length without affecting total sarcomere length[34]. The latter is well in line with the reduced maximal force we show in single cardiomyocytes of fibulin-4[R/R] mice. Cofilin-mediated impairment of cytoskeletal organization and sarcomere dysfunction as a result of reduced fibulin-4 expression would both contribute to reduced cardiomyocyte contraction[35].

Additionally, our study suggests that increased fibroblast content in fibulin-4[+/R] and fibulin-4[R/R] cardiac microtissues might contribute to reduced beating frequency. However, the observed differences in beating frequency in vitro were not evident in our in vivo measurements, implying compensatory mechanisms that might mask fibulin-4-related defects in vivo. Furthermore, our findings indicate that reduced fibulin-4 expression can directly affect cardiomyocytes, independent of fibulin-4-mediated elastogenesis, potentially leading to primary myocardial dysfunction. Next to cardiomyocytes, cardiac fibroblasts are also key players in the maintenance of normal cardiac function[36]. Elevated cardiac fibroblast content has been reported to reduce conduction velocity[37,38]. Despite the significant increase in fibroblast content in the microtissues of fibulin-4[+/R] and fibulin-4[R/R] cardiac cells, the cardiomyocyte-fibroblast ratio of all microtissues in this study is comparable to the cellular composition found in healthy neonatal murine hearts[39]. Moreover, the percentage of fibroblasts is almost identical in the fibulin-4[+/R] and fibulin-4[R/R] microtissues. In spite of that, the contractile force is not affected in the fibulin-4[+/R]-derived microtissues but significantly reduced in microtissues derived from fibulin-4[R/R] hearts. We

therefore conclude that the observed reduced contraction force is not caused by a difference percentage of fibroblasts but the result of reduced fibulin-4 expression. Yet, possibly the increased fibroblast content in the microtissues of fibulin-4$^{+/R}$ and fibulin-4$^{R/R}$ cardiac cells contributed to the reduced beating frequency is these microtissues. The fact that the differences in beating frequency were not found in our in vivo measurement might be explained by the absence of systemic mechanisms to control heart rate in vivo, including pacemaker cells and adrenergic stimulation, in the in vitro cardiac tissues. This also indicates that the microtissue model system is capable of unmasking fibulin-4 related defects that might be compensated in vivo. Since the microtissues were cultured for a period of only 2 days, most likely very little endogenous ECM has been produced. Consequently, it is unlikely that impaired elastic fiber assembly affected the adhesion and contractility of the cardiomyocytes at this early time point, showing that fibulin-4 deficiency can cause a primary defect in cardiomyocytes that is independent of fibulin-4 mediated elastogenesis.

In fibulin-4$^{+/R}$ mice, survival post-sTAC was significantly impacted compared to mTAC. Although an increase in fibulin-4 expression was observed in mTAC-treated fibulin-4$^{+/R}$ mice, this response was absent after sTAC. The heterogeneous fibulin-4 expression in response to mTAC implies varied susceptibility to cardiac pressure overload due to variations in fibulin-4 levels within cardiac tissue. However, further investigations with earlier time points are necessary to establish a correlation between fibulin-4 expression and cardiac failure. Employing cardiomyocytes from human pluripotent stem cells unaffected by aortic disease, lentiviral knockdown of fibulin-4 resulted in increased expression of cardiac stress-related genes and increased cell area. This supports the notion that reduced cardiac fibulin-4 levels lead to cardiomyocyte pathology independent of concurrent aorta-related factors.

Fibulin-4 plays a crucial role in maintaining cardiac myocyte function and homeostasis through its influence on the extracellular matrix (ECM). In cardiac myocytes, fibulin-4 is integral to the structural and functional integrity of the myocardium. Specifically, fibulin-4 is involved in the regulation of the ECM surrounding cardiomyocytes, as evidenced by its increased expression in response to pressure overload (TAC). This suggests that fibulin-4 plays a vital role in ECM remodeling, which is essential for preserving myocardial function under stress conditions. The deficiency of fibulin-4 leads to decreased contractile force and altered mechanical properties of cardiomyocytes, indicating its critical role in optimal myocyte contractility and function. This effect is likely mediated through fibulin-4's interactions with ECM components that are crucial for force transmission and cellular signaling within cardiomyocytes. Other connective tissue disorders, such as fibrillin-1 deficiency in Marfan Syndrome, could affect TGF-β signaling[40]. In disagreement with the observations in Marfan Syndrome mice, we did not find any alterations in ERK and Smad2 phosphorylation status.

Additionally, reduced fibulin-4 expression contributes to cardiomyocyte hypertrophy and exacerbates the pathological response to pressure overload. This observation implies that fibulin-4 modulates hypertrophic signaling pathways, potentially through its influence on ECM composition and mechanical stress sensing. Furthermore, the absence of fibulin-4 impairs ECM remodeling processes, as indicated by increased fibrosis and altered ECM composition in the heart. This disruption in ECM integrity contributes to impaired cardiac function and an increased susceptibility to stress-induced damage. Although non-myocyte cells such as fibroblasts also express fibulin-4, the specific effects on cardiomyocyte function underscore the importance of myocyte-specific roles for this protein. This is supported by the functional differences observed in cardiomyocytes from fibulin-4-deficient models, despite the presence of fibulin-4 in non-myocyte cells.

In summary, our study identifies fibulin-4 as a critical gene expressed in cardiomyocytes that is pivotal for mitigating cardiac dysfunction and pathological remodeling induced by pressure overload. This delineates fibulin-4's crucial role, beyond its involvement in elastic fiber formation, in maintaining cardiac function through regulation of myocyte contractility and modulation of the hypertrophic response to pressure overload. The mechanistic details of these roles likely involve interactions with ECM components and signaling pathways that influence both the structural and functional aspects of the myocardium. These insights hold potential significance for preventing and managing cardiac diseases in individuals with elastic fiber disorders. Given the improved medical and surgical treatments for aortic valvular pathology, our findings suggest the necessity to focus on cardiac care in patients with connective tissue disorders to prevent potential cardiac pathologies, supplementing the current emphasis on preventing aortic root expansion.

## Methods
### Experimental animals
We previously generated a fibulin-4 allele with reduced expression by transcriptional interference through placement of a TKneo targeting construct in the downstream Mus81 gene; heterozygous fibulin-4$^{+/R}$ mice in a mixed C57Bl/6 J;129 Sv background were mated to obtain fibulin-4$^{+/+}$ and fibulin-4$^{R/R}$ littermates[15]. Systemic fibulin-4 expression is reduced 50% in fibulin-4$^{+/R}$ and 75% in fibulin-4$^{R/R}$ aortas[15]. In this study, 14-week-old male and female mice were used to conduct TAC. Mice were sacrificed 28 days post-TAC. Myocardial fibulin-4 protein levels were determined through western blot analysis. Animals were housed at the Animal Resource Center (Erasmus University Medical Centre), and all experiments were conducted in accordance with the Principles of Laboratory Animal Care and the guidelines approved by the Dutch Ethical Committee in full compliance with European legislation.

Mice included in the experiment were C57BL/6J;129 Sv, aged 8–12 weeks, with a baseline body weight within 10% of the cohort average. Exclusion criteria included animals with pre-existing cardiac abnormalities, unexpected health issues (e.g., severe weight loss >15%, infections, or distress), or unsuccessful TAC surgery (evidenced by Doppler measurement confirming no significant pressure gradient across the aortic constriction). Criteria were established a priori. Mice were assigned to either the TAC or sham surgery group in an alternating manner based on their order of entry into the experiment. No formal random number generator was used, but care was taken to ensure an approximately equal distribution of body weight and age between groups. To reduce potential confounders, all mice were housed under identical conditions (same room, temperature, and light-dark cycle), and littermates were evenly distributed between groups. Surgical procedures were performed in a randomized order to prevent time-of-day effects. Measurements (e.g., echocardiography, Doppler, or histological analysis) were performed in a blinded manner, ensuring that the investigator was unaware of group assignments during data collection and analysis. I.P. was aware of group allocation during allocation. E.D.D. and N.V. were aware of group allocation during allocation, the conduct of the experiment, the outcome assessment, and the data analysis. M.S. was aware of group allocation during analysis. In the TAC group, 2 mice were excluded due to unsuccessful surgery (determined by Doppler ultrasound). In the sham group, 0 mice were excluded due to perioperative complications. No data points were excluded during analysis unless technical errors (e.g., imaging artifacts, signal dropout) were identified.

### Induction of TAC and in vivo measurements
14-week-old male and female fibulin-4$^{+/+}$ and fibulin-4$^{+/R}$ mice (total n = 59) were subjected to severe TAC (sTAC) (fibulin-4$^{+/+}$ n = 5, fibulin-4$^{+/R}$ n = 5), or mild TAC (mTAC) (fibulin-4$^{+/+}$ n = 11, fibulin-4$^{+/R}$ n = 17) using a 27 G or 25 G needle, respectively, or a sham operation (fibulin-4$^{+/+}$ n = 11, fibulin-4$^{+/R}$ n = 10)[41]. Echocardiography and hemodynamic measurements were performed 4 weeks after surgery. LV pressure and aortic pressure proximal and distal to the stenosis were measured. The systolic pressure gradient over the stenosis was used as a measure for stenosis severity. Subsequently, LVW, RVW, lung weight as well as TL were determined, and LV tissue samples were stored for histological and molecular analysis.

## Echocardiographic and hemodynamic measurements

Echocardiographic and hemodynamic measurements were performed in 18-week-old fibulin-4$^{+/+}$ (n = 16) and fibulin-4$^{R/R}$ mice (n = 6) 4 weeks post-TAC (total n = 22). All mice were anesthetized with 2.5% isoflurane, intubated, and ventilated. Echocardiography of the ascending aorta and left ventricle (LV) was performed using a Vevo2100 (VisualSonics Inc., Toronto, Canada). LV lumen diameters and ejection fraction (EF) were obtained from M-Mode images, and pulse wave Doppler was used to visualize aortic regurgitation. Subsequently, aortic and LV pressure were measured. 15 Consecutive beats were averaged, and several indices of cardiac function were calculated from the average LV pressure signal. First, we calculated the maximum rate of rise of LV pressure (LV dP/dt$_{max}$), which is a measure of LV contractility. Comparably, the maximum rate of fall of LV pressure (LV dP/dt$_{min}$) can be used as a measure of LV relaxation. Because the LV dP/dt$_{max}$ is load sensitive and TAC mice have an increased afterload, we additionally calculated the less afterload-sensitive index of cardiac contractility; the rate of rise of LV pressure at a pressure of 40 mmHg (LV dP/dt$_{P40}$). Likewise, a less afterload-sensitive measure of cardiac diastolic function is the time constant of LVP decay (tau). Tau was computed from the following equation:

$$LVP(t) = LVP_0 \cdot e^{-t/tau}$$

in which LVP$_0$ = LVP at LV dP/dt$_{min}$, using data points during the isovolumic relaxation phase, starting at LVP$_0$ until LV pressure reached a value of 5 mmHg above LVEDP[41].

At the end of each experiment, LV, right ventricle (RV), and lung weights, as well as tibia lengths (TL), were determined, and LV tissue samples (not containing aorta or aortic valve tissue) were stored for histological and molecular analysis.

## Cardiac histology

Paraffin embedded LV tissue (total n = 24, n = 6 per group) was serially sectioned into 5-µm slices. LV sections were stained with Gomori's silver staining for determination of cardiomyocyte cross-sectional area. LV sections were stained with Picrosirius Red to assess collagen accumulation. Images were captured using commercial software (CLEMEX Vision and Visiopharm).

Additional heart sections were boiled in 100 mM Tris-HCl (pH 9.0) and 10 mM EDTA to promote antigen exposure and emerged in 3% H$_2$O$_2$ in methanol to inhibit endogenous peroxidase activity prior to pSmad2 and CTGF staining After blocking and 0.025% Triton treatment the slides were incubated with antibodies against phospho-Smad2 (Ser465-467, #3108, Cell Signaling) or CTGF (#GTX47807, GeneTex) and counterstained with hematoxylin.

HE stainings on cardiac tissue were done using the automated HE600 (Ventana, Roche). Trichome blue staining was performed on the automated Ventana special stains (Roche). In short, after deparrafinization 300 ul of Trich Bouins A was incubated for 32 min, after several special stains washing steps hematoxylin A + B was added for 12 min. Slides were washed and incubated with Trich Red for 8 min followed by incubation of Trich Mordant for 12 min. Trich Blue was added for 16 min followed by several washing steps, to finally be incubated with Trich Clarifier for 4 min. Slides were sealed using the coverslipping program in the HE600.

## Immunofluorescence staining

Four microliters thick tissue sections on extra adhesive glass slides (Leica, Biosystems) were processed in the Discovery Ultra instrument (Ventana, Roche). The "tissues" automated Discovery Universal protocol was used. Short pretreatment was performed with CC1 for 8 min (Ventana). One drop of DISC inhibitor (Ventana) was applied and incubated for 12 min. Next, Rb anti-Elastin (1:100, #ab307150, Abcam) or Rb anti-Fibulin-4 (1:100, provided by T. Sasaki[9]) antibody was incubated for 32 min at 37 °C followed by detection with omnimap anti-Rb HRP (Ventana) for 20 min and visualization with Cy5 for 8 min (Ventana). Slides were cleaned and coverslipped

with DAPI in Vectashield. Slides were scanned using the Zeiss Axioscanner 7.0 using 40x magnification.

iPSC-cardiomyocytes were fixed with 2% PFA for 15 min and washed, and permeabilized in PBS with 0.1% Triton X-100 for 2 × 10 min and then blocked in PBS with 0.5% BSA and 0.15% glycine for 30 min. Fixed cells were incubated overnight in fibulin-4 (1:250, provided by D. Reinhardt[42]) and α-actinin (1:250, #A7811, Sigma-Aldrich) antibodies. After washing in PBS with 0.1% Triton X-100 for 2 × 10 min, secondary antibodies (Molecular Probes) were incubated for 1 h. After another washing step, VectaShield with DAPI was added, and the slide was sealed with a glass coverslip. Images were taken using a Leica SP5 confocal microscope at 63x magnification. For measuring the cell size of iPSC-cardiomyocytes, cells were stained using a α-myosin heavy chain (10 µg/mL, clone MF20, #MAB4470, R&D Systems) fluorescently-labeled antibody.

## Plasma renin concentration

Plasma renin concentration (PRC) was determined (total n = 24, n = 6 per group) by enzyme-kinetic assay in the presence of excess angiotensinogen[43].

## Microtissue engineering and contractility analysis

After sacrificing 1- to 3-day-old neonatal mice and isolation of the hearts, each heart was individually incubated in trypsin overnight. A piece of the tail from each mouse was digested for genotype determination. The next day, hearts were pooled according to their genotype, and cells were isolated and cultured[23,24]. Cardiac microtissues were seeded in a previously developed µTUG system with uniaxial constraints[23]. At day 2 and day 4 of culture, movies of the microtissues were recorded using a high-speed camera on a Zeiss Observer microscope. Dynamic contraction force and beating frequency were calculated from the micropost displacements over time[23]. Subsequently, microtissues were stained for α-actinin (1:800, #A7811, Sigma-Aldrich) and vimentin (1:400, #5741T, Cell Signaling), using fluorescent antibodies to determine sarcomere length and fibroblast content.

## Isometric force measurements

The force generating capacity of single membrane-permeabilized cardiomyocytes from fibulin-4$^{+/+}$, fibulin-4$^{+/R}$, and fibulin-4$^{R/R}$ mice was assessed[41]. In short, myocytes from 3 LV sections per group (2–4 cells per section) were isolated and all membranes removed. Isometric force measurements were performed in activation solutions with different calcium concentrations to determine myofilament maximal force (F$_{max}$), calcium-sensitivity (pCa$_{50}$), and passive force (F$_{pas}$).

## Mouse LV mRNA analysis

mRNA expression analyses of LV samples (n = 5 per group) were performed using real-time fluorescence assessment of SYBR Green Primer sets for atrial natriuretic peptide (ANP), brain natriuretic peptide (BNP), α-skeletal muscle actin (α-SKA), sarcoplasmic reticulum Ca$^{2+}$ ATPase (SERCA2a) and connective tissue growth factor (CTGF) (Integrated DNA Technologies, Coralville, USA) (primer sets are listed in Table S2). mRNA levels were corrected for the housekeeping gene β-actin and normalized to sham Fibulin-4$^{+/+}$.

## Western blotting

LV tissue samples (n = 3–4 per group) were used for immunoblotting with antibodies to fibulin-4[44], phospho- Mothers against decapentaplegic homolog 2 (pSMAD2) (#AB3849, Millipore), SMAD2 (#5339S, Cell Signaling), phospho-extracellular signal-regulated protein kinases 1 and 2 (pERK1/2) (Thr202/Tyr204, #4370, Cell Signaling), ERK1/2 (#4696, Cell Signaling), phospho-focal adhesion kinase (pFAK) (Tyr397, # PA5-85602, Invitrogen), FAK (#05-537, Milipore) and SERCA2a (#ab3625, Abcam). pERK1/2, ERK1/2, pFAK, and FAK protein levels were corrected using corresponding total protein staining (Licor), after which pERK1/2 / ERK1/2 and pFAK/FAK levels were calculated. Raw blot images can be found in Figs. S4–S7.

## Cell culture and knockdown of fibulin-4 by lentivirus-mediated shRNA

Two lentiviral vectors containing shRNAmirs against Fibulin-4 as well as a non-silencing control shRNAmirs (Thermo Scientific, Huntsville, USA) were used to generate lentivirus. To inhibit endogenous fibulin-4, human iPSC-derived cardiomyocytes (iCell cardiomyocytes, CDI, Madison, WI, USA) were infected with the lentiviral shRNAmir constructs, achieving an infection rate of ~90%. Four days after infection, the cells were harvested or fixed and stained with fluorescently-conjugated antibodies to α-myosin heavy chain (α-MHC) (clone MF20, #MAB4470, R&D) and a polyclonal fibulin-4 antibody that was characterized previously[45]. Additionally, the purity of iPSC-derived cardiomyocyte-cultures was determined as the percentage of α-MHC-stained cells. At the time of collection, the cultures contained ~98% α-MHC positive cells demonstrating that only 2% of the total cell population consisted of non-cardiomyocytes.

## Gene expression analysis in human iPSC-derived cardiomyocytes

Levels of human iPSC-derived cardiomyocyte mRNA were analyzed by real-time PCR using Taqman primers against fibulin-4 (*EFEMP2*), connective tissue growth factor (*CCN2*), plasminogen activator inhibitor type 1 (*SERPINE1*), natriuretic peptide A (*NPPA*), phospholamban (*PLN*), myosine heavy chain 7 (*MYH7*) and myosine heavy chain 6 (*MYH6*) (Applied Biosystems, Foster City, CA, USA) and corrected for the housekeeping gene Large Ribosomal Protein subunit P0 (Table S3).

## Statistics and reproducibility

Sample size was determined based on an alpha of 0.05 and a power of 80% using a two-tailed test. A difference of 25% was expected between wild type and fibulin-4$^{+/R}$ littermates with a standard deviation of 18%. Based on this, we calculated the sample size to be N = $2*[(1.96 + 0.84)^2/(25/18)^2]$ = 8.1 = 9 animals per group. We expected a 35% dropout due to heart failure as a result of the surgery, and a 5% dropout during surgery due to technical reasons. To keep 9 animals in the TAC groups, the size for these groups was set to 15 animals. In total, 354 animals were required (sham; 9 animals for 7 timepoints = 63 wild type and 63 fibulin-4$^{+/R}$ animals, TAC; 15 animals for 7 timepoints = 105 wild type and 105 fibulin-4$^{+/R}$ animals, no surgery; 9 animals for 2 timepoints = 18 fibulin-4$^{R/R}$ animals).

A *t* test was applied to compare fibulin-4$^{+/+}$ with fibulin-4$^{R/R}$ mice, and one-way ANOVA was performed to evaluate results from iPSC-derived cardiomyocytes. Statistical comparison of fibulin-4$^{+/+}$ and fibulin-4$^{+/R}$ with or without mTAC was performed using a two-way ANOVA, followed by a Dunnett post-test, when appropriate. Survival was analyzed by Kaplan–Meier method and log-rank (Mantel-Cox) test. Sample sizes (number of mice) are specified in the figure legends. Comparisons of a microtissue parameter between the three phenotypes were made using a non-parametric Kruskal–Wallis test with Dunns post-test. A *p* value < 0.05 (two-tailed) was considered statistically significant. Data are presented as the mean ± standard error. All statistical analyses were performed using GraphPad Prism 9 (version 9.5.1).

## Reporting summary

Further information on research design is available in the Nature Portfolio Reporting Summary linked to this article.

## Data availability

Raw and processed data are included in the supplementary data. All other data are available from the corresponding author.

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

## Acknowledgements
The authors would like to thank Ingrid van den Berg–Garrelds for her measurements of the plasma renin concentrations in fibulin-4$^{+/+}$ and fibulin-4$^{+/R}$ mice. This work was supported by the "UNESCO-L'Oréal for Woman in Science Fellowship" (E.D.D.), the Netherlands Institute for Regenerative medicine (FES0908, A.C.C.S., N.A.M.B and C.V.C.B.), Genetic Aortic Disorders Association Canada and Natural Sciences and Engineering Research Council of Canada (D.P.R.), the British Heart Foundation Centre of Research Excellence and British Heart Foundation Simon Marks Chair in Regenerative Cardiology (M.D.S.), the "Lijf en Leven"-grant (2008): "Early detection and diagnosis of aneurysms and heart valve abnormalities" (J.E.) and Dilating versus Stenosing arterial disease (2011) (E.D.D., I.P., and J.E.), the Heart-CHIP II Health~Holland project (grant number EMCLSH19005, M.S.), the Dutch CardioVascular Alliance, an initiative with financial support of the Dutch Heart Foundation (2017B018-ARENA-PRIME & 2021B008-RECON-NEXT, D.J.D.), and TKI-LSH grant Quantitative in vivo imaging of heart failure (HF-Image, LSHM18002, N.V.).

## Author contributions
J.E. and I.P. conceived and supervised the project. J.V., C.V.C.B, J.H.T., A.H.J.D., D.J.D., and M.D.S. supervised the project. E.D.D. and M.S. wrote the manuscript and contributed equally to this work. E.D.D., M.S., N.V., L.R., T.P.P.B., L.R.F., A.C.C.S., N.A.M.B., and N.B. performed the experiments. C.M.H., T.S., and D.P.R. provided materials and resources. All authors contributed to the article and approved the final version for submission.

## Competing interests
The authors declare no competing interests.

## Additional information

[1]Department of Molecular Genetics, Erasmus University Medical Center, Rotterdam, the Netherlands. [2]Division of Experimental Cardiology, Department of Cardiology, Erasmus University Medical Center, Rotterdam, the Netherlands. [3]Division of Pharmacology, Department of Internal Medicine, Erasmus University Medical Center, Rotterdam, the Netherlands. [4]Department of Vascular Surgery, Cardiovascular Institute, Erasmus University Medical Center, Rotterdam, the Netherlands. [5]Department of Pathology and Clinical Bioinformatics, Erasmus University Medical Center, Rotterdam, the Netherlands. [6]British Heart Foundation Centre of Research Excellence, National Heart and Lung Institute, Imperial College London, London, UK. [7]Department of Biomedical Engineering, and Institute for Complex Molecular Systems, Eindhoven University of Technology, Eindhoven, the Netherlands. [8]Department of Physiology, VU University Medical Center, Institute for Cardiovascular Research (ICaR-VU), Amsterdam, the Netherlands. [9]Division of Nephrology, Department of Pediatrics, Washington University School of Medicine, St. Louis, MO, USA. [10]Department of Pharmacology, Faculty of Medicine, Oita University, Oita, Japan. [11]Faculty of Medicine and Health Sciences, McGill University, Montreal, QC, Canada. [12]Faculty of Dentistry and Oral Health Sciences, McGill University, Montreal, QC, Canada. [13]Department of Radiotherapy, Erasmus University Medical Center, Rotterdam, the Netherlands. [14]These authors contributed equally: E. D. van Deel, M. Snelders. ✉e-mail: j.essers@erasmusmc.nl

