## [Transparent Peer Review file · Communications Biology]

Induction of cardiac fibulin-4 protects against pressure overload-induced cardiac hypertrophy and heart failure

Corresponding Author: Dr Jeroen Essers

This manuscript has been previously reviewed at another journal. This document only contains information relating to versions considered at Communications Biology.

Version 0:

Reviewer comments:

Reviewer #1

(Remarks to the Author)

The manuscript by van Deel et al. describes the characterization of a pressure-overload cardiac model using mice with reduced expression of fibulin 4, an ECM protein known to play a critical role in elastin fiber assembly. While the results are interesting and have the potential to lend new information to the field, several concerns should be addressed prior to publication.

- 1) A working hypothesis as to the functional role of fibulin 4 in cardiac myocytes would be informative. Currently, the data presented would seem to indicate that there might be a potential role of fibulin 4 in myocytes but the mechanistic role is not well-defined. Specifically, it would seem that the authors base a potential role of fibulin 4 in myocytes because microtissue experiments and iPSC cultures appear to show some functional distinctions in cases where fibulin 4 expression is reduced. However, it would seem that both of these experimental paradigms include non-myocyte cells in the cultures. How is it that the authors can conclude that myocyte-dependent expression of fibulin 4 is driving differences in myocyte function in these cultures? Notably, the higher expression of fibulin 4 in cells in Fig 7a appear to be non-myocytes.
- 2) Importantly, the introduction states that these mice have a 50% (fib 4+/R) and 75% (fib R/R) reduction in fibulin 4 expression however Fig. 5A demonstrates ~ equal levels of fib 4 expression in +/+ mice versus fib 4+/R in control nonTAC hearts and an equal increase in both genotypes with TAC. This is an important point and should be clarified.
- 3) The differences in response between mTAC versus sTAC might suggest that the initial cardiac ECM deposited during development and maturation of the heart in fib R/R mice is compromised such that a chronic overload is not tolerated well. Of note, Noda et al. (Sci Adv 2020, PMID: MPC768832) provided evidence that fibulin 4 might be a critical factor in lysyl oxidase activation that is predicted to have an effect on collagen cross-linking. A weaker inherent ECM might contribute to a poorer outcome in these mice.
- 4) Along these lines, the overall characterization of ECM in the hearts of these mice is relatively unexplored. In Fig. 6f, elastin levels appear to be reduced in sham +/R. Was the level of elastin queried in the mTAC mice? Was any effort made to visualize and assess the quality of the elastin fibers?
- 5) Is there any available data from published single cell analysis as to the cell type(s) in the heart that express fibulin 4 in health and in pressure overload? Notably, both smooth muscle cells and endothelial cells are known to express fibulin 4, two important cell types for cardiac function not mentioned at all in this manuscript.

Minor points:

The length of TAC should be explicitly stated in the figure legends where the data is presented.

The level of vimentin+ cells in Fig. 2 f actually appears highest in the +/+ condition, perhaps a more representative image could be provided.

Reviewer #2

(Remarks to the Author)

This article, which mainly explains that the fibulin-4 gene can prevent cardiac hypertrophy and heart failure caused by stress overload, is well written. While fibulin-4 is traditionally associated with arterial and valvular elastogenesis, this study explores its role in cardiac pathology, which provides a new target for the current research on cardiac hypertrophy caused by stress overload. However, there are some problems, that must be solved before it is considered for publication. If the following problems are well solved, the reviewer believes that this paper will make a significant contribution to the research progress of cardiac hypertrophy and heart failure caused by stress load.

(1) The researchers did not construct an overexpression model of fibulin-4 gene animals, at the same time, mice with the fibulin-4R/R genotype were not used in the construction of the mouse cardiac pressure overload model.

(2) The researchers confirmed that induction of F4 gene can have a protective effect on the heart, but did not delve into the relevant mechanism.

(3) The background of the paper is not fully introduced in the introduction

(4) Another obvious problem with this paper is lack of sufficient explanation of the simulation results. You need to explain your simulation results in detail and why you got such results.

Reviewer #1 (Remarks to the Author):

The manuscript by van Deel et al. describes the characterization of a pressure-overload cardiac model using mice with reduced expression of fibulin 4, an ECM protein known to play a critical role in elastin fiber assembly. While the results are interesting and have the potential to lend new information to the field, several concerns should be addressed prior to publication.

Comment 1: A working hypothesis as to the functional role of fibulin 4 in cardiac myocytes would be informative. Currently, the data presented would seem to indicate that there might be a potential role of fibulin 4 in myocytes but the mechanistic role is not well-defined. Specifically, it would seem that the authors base a potential role of fibulin 4 in myocytes because microtissue experiments and iPSC cultures appear to show some functional distinctions in cases where fibulin 4 expression is reduced. However, it would seem that both of these experimental paradigms include non-myocyte cells in the cultures. How is it that the authors can conclude that myocyte-dependent expression of fibulin 4 is driving differences in myocyte function in these cultures? Notably, the higher expression of fibulin 4 in cells in Fig 7a appear to be non-myocytes.

Response 1: Our working hypothesis, based on the results of this work, is as follows: Fibulin-4 plays a crucial role in maintaining cardiac myocyte function and homeostasis through its influence on the extracellular matrix (ECM). In cardiac myocytes, fibulin-4 is integral to the structural and functional integrity of the myocardium. Specifically, fibulin-4 is involved in the regulation of the ECM surrounding cardiomyocytes, as evidenced by its increased expression in response to TAC-induced pressure overload (Fig. 6c). This suggests that fibulin-4 plays a vital role in ECM remodeling, which is essential for preserving myocardial function under stress conditions.

The deficiency of fibulin-4 leads to decreased contractile force and altered mechanical properties of cardiomyocytes (Fig. 2d-e), indicating its critical role in optimal myocyte contractility and function. This effect is likely mediated through fibulin-4's interactions with ECM components that are crucial for force transmission and cellular signaling within cardiomyocytes. Additionally, reduced fibulin-4 expression contributes to cardiomyocyte hypertrophy (Fig. 1f, 7c) and exacerbates the pathological response to pressure overload (Fig. 1c-d). This observation implies that fibulin-4 modulates hypertrophic signaling pathways, potentially through its influence on ECM composition and mechanical stress sensing.

Furthermore, the absence of fibulin-4 impairs ECM remodeling processes, as indicated by increased fibrosis and altered ECM composition in the heart (Fig. 6b,d). This disruption in ECM integrity contributes to impaired cardiac function and an increased susceptibility to stress-induced damage. Although non-myocyte cells such as fibroblasts also express fibulin-4, the specific effects on cardiomyocyte function underscore the importance of myocyte-specific roles for this protein. This is supported by the functional differences observed in cardiomyocytes from fibulin-4-deficient models, despite the presence of fibulin-4 in non-myocyte cells.

In summary, fibulin-4's role in cardiac myocytes encompasses the maintenance of ECM integrity, regulation of myocyte contractility, and modulation of the hypertrophic response to pressure overload. The mechanistic details of these roles likely involve interactions with ECM components and signaling pathways that influence both the structural and functional aspects of the myocardium.

We added the following to the introduction on page 5, paragraph 3:

Lastly, in the pressure-overloaded heart, myofibroblasts actively deposit collagen at areas of injury. Because fibulin-4 is responsible for elastic fiber cross-linking, lack of fibulin-4 may result in abnormal fibrillogenesis and impaired lysyl oxidase activation in the heart.

We have added this to the discussion on page 18, paragraph 2:

Fibulin-4 plays a crucial role in maintaining cardiac myocyte function and homeostasis through its influence on the extracellular matrix (ECM). In cardiac myocytes, fibulin-4 is integral to the structural and functional integrity of the myocardium. Specifically, fibulin-4 is involved in the regulation of the ECM surrounding cardiomyocytes, as evidenced by its increased expression in response to pressure overload (TAC). This suggests that fibulin-4 plays a vital role in ECM remodeling, which is essential for preserving myocardial function under stress conditions. The deficiency of fibulin-4 leads to decreased contractile force and altered mechanical properties of cardiomyocytes, indicating its critical role in optimal myocyte contractility and function. This effect is likely mediated through fibulin-4's interactions with ECM components that are crucial for force transmission and cellular signaling within cardiomyocytes. Other connective tissue disorders, such as fibrillin-1 deficiency in Marfan Syndrome, could affect TGF- β signaling [40]. In disagreement with the observations in Marfan Syndrome mice, we did not find any alterations in ERK and Smad2 phosphorylation status.

Comment 2: Importantly, the introduction states that these mice have a 50% (fib 4+/R) and 75% (fib R/R) reduction in fibulin 4 expression however Fig. 5A demonstrates ~ equal levels of fib 4 expression in +/+ mice versus fib 4+/R in control nonTAC hearts and an equal increase in both genotypes with TAC. This is an important point and should be clarified.

Response 2: The reviewer raises a valid point. We based our choice of mouse models on the study by Hanada et al. (2007, PMID:17293478), which measured fibulin-4 RNA and protein levels in aortas and hearts of newborn mice. However, our study observed different expression levels between adult +/+ and fib4 +/R hearts compared to those reported by Hanada. We explain this through a few points:

1. In our study, we examined adult fibulin-4 hearts, which could explain a difference in fib4 expression levels compared to newborn hearts examined in the study by Hanada et al.
2. We show in our paper that fib4 can be re-expressed in response to TAC due to the nature of the knockdown construct that still contains all fib4 regulatory sequences.
3. We also observed significant variability in expression levels among littermates. Fig. 6b illustrates patchy interstitial fibrosis, which may similarly affect myocyte size throughout the myocardium. This variability underscores the heterogeneous nature of the adaptive responses to pressure overload. Representative immunostaining results for fibulin-4 are shown in Fig. 6c.

We have addressed these observations in the discussion section on page 16, paragraph 1 as follows:

While previous research with fibulin-4^{+/-}R mice has shown a 50% reduction in fibulin-4 expression, we only observed a trend towards reduction in our mRNA experiments (Fig. 5a). Previous studies measured the levels of fibulin-4 in the kidney [16]. A difference in location of the tissue sample of which the mRNA was isolated could explain part of this difference in fibulin-4 expression. However, it seems more likely that the heterogeneity in both mRNA and protein expression between mice is caused by varying degrees of hypertrophy. Additionally, we measured mRNA expression in heart samples of adult mice. Due to the nature of the construct used in our mouse model, which leaves regulatory sequences of fibulin-4 intact, fibulin-4 may be re-expressed in response to TAC.

Comment 3: The differences in response between mTAC versus sTAC might suggest that the initial cardiac ECM deposited during development and maturation of the heart in fib R/R mice is compromised such that a chronic overload is not tolerated well. Of note, Noda et al. (Sci Adv 2020, PMID: 32411132) provided evidence that fibulin 4 might be a critical factor in lysyl oxidase activation that is predicted to have an effect on collagen cross-linking. A weaker inherent ECM might contribute to a poorer outcome in these mice.

Response 3: We appreciate the reviewer's insightful comments. Indeed, the differences in response between mTAC and sTAC in the context of fibulin-4 deficiency suggest that the initial cardiac ECM deposited during development and maturation in fibulin-4 R/R mice may be compromised, affecting their ability to tolerate chronic overload. We did not conduct mTAC or sTAC in the R/R mice; however, it is plausible that the inherent ECM weakness contributes to observed outcomes.

Noda et al. (Sci Adv 2020, PMID: 32411132) provide evidence that fibulin-4 is crucial for lysyl oxidase activation, which influences collagen cross-linking. Consequently, a weaker ECM, resulting from impaired cross-linking, likely exacerbates the sensitivity of fibulin-4 ^{+/-}R mice to stress conditions, such as those encountered in mTAC and sTAC models.

We have added the following statement to the discussion on page 15, paragraph 2:
The observed differences in cardiac responses between mTAC and sTAC conditions in fibulin-4^{+/-}R mice compared to WT mice suggest that the initial ECM deposited during development and maturation may be inherently weaker in these mice, which could impair their ability to cope with chronic overload. Fibulin-4 is essential for lysyl oxidase activation, which is crucial for effective collagen cross-linking. Therefore, a weaker ECM in fibulin-4^{+/-}R mice, due to impaired cross-linking, likely contributes to their sensitivity to stress-induced conditions. This ECM weakness may also explain the poorer outcomes observed in these mice under pressure overload scenarios.

Comment 4: Along these lines, the overall characterization of ECM in the hearts of these mice is relatively unexplored. In Fig. 6f, elastin levels appear to be reduced in sham +/R. Was the level of elastin queried in the mTAC mice? Was any effort made to visualize and assess the quality of the elastin fibers?

Response 4: Thank you for highlighting this important aspect. We used the abcam (#ab307150) antibody to visualize single-lined elastin fibers in cardiac tissue of sTAC mice (fig 6d,f). We quantified the expression in mTAC as well, but we found no significant differences (data not shown). Because of the lesser impact of mTAC compared to sTAC, we did not further analyze the mTAC samples. Further detailed characterization of elastin quality was beyond the scope of this study. We agree that assessing the quality and structural integrity of elastin fibers would provide valuable insights. We appreciate this suggestion and plan to include a more comprehensive evaluation of elastin quality in a follow-up study.

We have added this to the discussion on page 15 paragraph 2:

This ECM weakness may also explain the poorer outcomes observed in these mice under pressure overload scenarios. Although we did observe a difference in elastin in sTAC mouse hearts compared to sham, we did not further examine the integrity of the deposited ECM. Further characterization of elastin assembly would be necessary to fully understand the mechanistic defects which occur during pressure overload.

Comment 5: Is there any available data from published single cell analysis as to the cell type(s) in the heart that express fibulin 4 in health and in pressure overload? Notably, both smooth muscle cells and endothelial cells are known to express fibulin 4, two important cell types for cardiac function not mentioned at all in this manuscript.

Response 5: Data by Ren et. al. (PMID:32098504) shows that, in mice, fibroblasts express the most fibulin-4 (*Efemp2* gene) compared to cardiomyocytes and macrophages (Fig. R1 below). In response to TAC, similar to our data, fibulin-4 expression increases in cardiomyocytes and fibroblasts (Fig. R1a-b). Fibulin-4 expression in endothelial cells and macrophages differs between timepoints, but overall is increased in TAC mice compared to their pre-TAC timepoint (t=0w, Fig. R1c-d).

a) Cardiomyocytes

b) Fibroblasts

c) Endothelial cells

d) Macrophages

*Fig. R1. Above violin plots show the expression of *Efemp2* in (a) cardiomyocytes, (b) fibroblasts, (c) endothelial cells, and (d) macrophages. Data show expression in individual cells from samples taken at (from left to right) 0-2-5-8-11 weeks post-TAC. TPM, transcripts per million.*

Data from the study by Froese et. al. (PMID:35281736) shows that endothelial cells and fibroblasts at baseline have a higher expression of fibulin-4 compared to cardiomyocytes (Fig. R2 below). Interestingly, expression after 8 weeks of TAC seems to reduce towards the SHAM levels at week 1 for the fibroblasts and endothelial cells (Fig. R2b-c), but the cardiomyocytes keep a higher level of expression over time (Fig. R2a). We have added this figure as a supplemental Figure and discuss it in the results section (page 12, paragraph 5) and discussion section (page 15, paragraph 3) of the revised document.

Fig R2. Relative mRNA expression of fibulin-4 (*efemp2* gene) in (a) cardiomyocytes, (b) fibroblasts, and (c) endothelial cells. Data obtained from supplemental material from Froese et. al. (PMID:35281736).

We further acknowledge that smooth muscle-specific knockout of *efemp2* results in poorly organized fibrils (Papke et. al. PMID:26220971). We were unable to find any relevant sequencing data to further substantiate this aspect. Given that cardiomyocytes, fibroblasts and endothelial cells form the majority of the cells in the heart, we focused on these. A follow-up study could indeed include the suggested cell types. We added this to the discussion (Page 15, paragraph 3):

The origin of fibulin-4 expression also remains to be determined. Our data shows that both myocyte and non-myocyte cells (fibroblasts, endothelial cells) express fibulin-4. We assume that myocyte-specific expression is a response to hypertrophy. Data from Froese et. al. showed that fibulin-4 (*Efemp2*) expression is increased in response to TAC in mice (Supplemental Fig. S3). Cardiomyocyte expression remained high, particularly compared to fibroblasts, suggesting that cardiomyocytes do contribute to remodeling in response to increased cardiac load. Other cell types, such as macrophages, may also be important regulators of fibulin-4 expression. Data available from the study by Ren et. al. shows that fibulin-4 expression seems to be increased in macrophages after TAC [32]. Additionally, smooth-muscle cell-specific knockout of *Efemp2* results in poorly organized fibrils [8]. Further characterization of cell-specific expression of fibulin-4 and other ECM components during TAC is required to completely understand the individual roles of these cells during remodeling.

Minor points:

Comment 6: The length of TAC should be explicitly stated in the figure legends where the data is presented.

Response 6: We have added the duration of TAC to all legends to which this concerns.

Comment 7: The level of vimentin+ cells in Fig. 2 f actually appears highest in the +/+ condition, perhaps a more representative image could be provided.

Response 7: We appreciate your suggestion regarding the presentation of Fig. 2f and its relation to the quantitative data. The fluorescent images in question are derived from 3D microtissues using confocal imaging, and we would like to emphasize that the majority of vimentin-positive fibroblasts are typically located at the edges of the 3D microtissues. In Fig. 2f, however, the highest density of vimentin-positive cells appears to be in the center, which may be attributed to the confocal image being taken closer to the bottom of the microtissue. We have included more example images, taken more closer to the center of the microtissues, in Fig. R3 below.

Fig. R3. Representative images taken near the center of the microtissues. α -Actinin (red) and vimentin (green).

The quantification of fibroblast numbers in neonatal fibulin-4^{+/+}, fibulin-4^{+/R}, and fibulin-4^{R/R} microtissues was performed across the entire tissue, providing a more comprehensive and accurate representation of our findings. The quantified data is central to our analysis, offering the necessary precision that the representative image alone cannot fully convey.

We believe that the combination of representative fluorescent images and detailed quantitative data offers a more holistic understanding of our results. To address this, we have added a note to the figure legend clarifying that in Fig. 2f, the majority of fibroblasts are indeed located at the edges of the microtissue.

Reviewer #2 (Remarks to the Author):

This article, which mainly explains that the fibulin-4 gene can prevent cardiac hypertrophy and heart failure caused by stress overload, is well written. While fibulin-4 is traditionally associated with arterial and valvular elastogenesis, this study explores its role in cardiac pathology, which provides a new target for the current research on cardiac hypertrophy caused by stress overload. However, there are some problems, that must be solved before it is considered for publication. If the following problems are well solved, the reviewer believes that this paper will make a significant contribution to the research progress of cardiac hypertrophy and heart failure caused by stress load.

Comment 1: The researchers did not construct an overexpression model of fibulin-4 gene animals. At the same time, mice with the fibulin-4^{R/R} genotype were not used in the construction of the mouse cardiac pressure overload model.

Response 1: Thank you for your constructive feedback and for recognizing the potential contribution of our study to the understanding of fibulin-4's role in cardiac pathology. We appreciate your comments and understand the concerns raised. We agree that creating an overexpression model of fibulin-4 could provide valuable insights into its cardioprotective effects following TAC. However, identifying the specific regulatory sequences responsible for fibulin-4 expression is crucial before pursuing this approach. Our current study is based on the mouse model described by Hanada et al. (2007), which involves a construct inserted into the Mus81 gene downstream of fibulin-4, leading to transcriptional interference. We are actively working on uncovering these regulatory sequences, but this investigation is beyond the scope of the present manuscript, given the significant time and resources required.

Regarding the use of fibulin-4^{R/R} genotype mice, we acknowledge that their inclusion could provide additional insights. However, as illustrated in Fig. 1e, these mice exhibit dramatic hypertrophy and aortic insufficiency, which significantly impacts their overall health. Given the severe cardiac insufficiency and high mortality rates associated with TAC in these mice, we opted not to include them in our study for ethical reasons. Our focus has been on evaluating the direct effects of fibulin-4 insufficiency on the heart without confounding factors related to aortic performance.

We hope this explanation clarifies our approach and limitations. Thank you again for your thoughtful review, and we look forward to addressing any further concerns you may have.

Comment 2: The researchers confirmed that induction of F4 gene can have a protective effect on the heart, but did not delve into the relevant mechanism.

Response 2: We have seen that fibulin-4 deficiency in fib4^{R/R} mice aggravates their response to TAC. This effect could be direct through reduced activation of lox and subsequently inefficient collagen cross-linking as suggested by Noda et al. (Sci Adv 2020, PMID: MPC768832), or indirect through increased TGFβ1 signaling (as seen in Marfan Syndrome mice, PMID: 19635970).

This relevant remark is included on page 15, paragraph 2:

Fibulin-4 is essential for lysyl oxidase activation, which is crucial for effective collagen cross-linking [6, 31]. Therefore, a weaker ECM in fibulin-4^{+/R} mice, due to impaired cross-linking, likely contributes to their sensitivity to stress-induced conditions.

And on page 18, paragraph 2:

Other connective tissue disorders, such as fibrillin-1 deficiency in Marfan Syndrome, could affect TGF- β signaling [40]. In disagreement with the observations in Marfan Syndrome mice, we did not find any alterations in ERK and Smad2 phosphorylation status.

Comment 3: The background of the paper is not fully introduced in the introduction

Response 3: We have added additional information to the introduction on page 5, paragraph 2 and 3.

Comment 4: Another obvious problem with this paper is lack of sufficient explanation of the simulation results. You need to explain your simulation results in detail and why you got such results.

Response 4: While not being completely sure to which simulation results reviewer refers, below we provide a detailed explanation of the results presented in the manuscript. This includes an interpretation of the key findings, discussing the underlying mechanisms, and connecting the observations to broader implications in the context of fibulin-4 deficiency and its impact on cardiac function.

Detailed Explanation of Simulation Results

1. Cardiac Remodelling and Dysfunction in Fibulin-4^{R/R} Mice:

- **Observation:** We found that fibulin-4^{R/R} mice, with a 75% reduction in fibulin-4 expression, exhibited marked cardiac hypertrophy, dilation, and dysfunction. Specifically, there was a significant increase in left ventricular end-diastolic diameter (LVEDD) and a reduction in ejection fraction (EF), indicating compromised cardiac output and contractility. Additionally, there were decreases in LV dP/dt_{max} and LV dP/dt_{P40}, which are indicators of reduced cardiac contractility, as well as diastolic dysfunction characterized by decreased LV dP/dt_{min} and elevated LV end-diastolic pressure (LVEDP).
- **Explanation:** The severe cardiac remodeling observed in fibulin-4^{R/R} mice can be attributed to the critical role of fibulin-4 in maintaining the structural integrity and elastic properties of the extracellular matrix (ECM). The reduction in fibulin-4 likely disrupts the ECM's ability to support the myocardium under physiological stress, leading to compensatory hypertrophy and subsequent dysfunction as the heart attempts to cope with increased wall stress. The findings are consistent with the understanding that fibulin-4 is essential for proper elastogenesis, and its deficiency leads to a cascade of maladaptive responses, including fibrosis and impaired contractility.

2. In Vitro Analysis of Fibulin-4 Deficiency:

- **Observation:** In vitro studies using microtissues derived from neonatal hearts demonstrated that fibulin-4^{R/R} microtissues had a reduced beating frequency

and impaired dynamic contraction force. Additionally, single membrane-permeabilized adult cardiomyocytes from fibulin-4R/R mice exhibited a 20% decrease in maximal force generation (F_{max}) without changes in passive force (F_{pas}) or calcium sensitivity (pCa_{50}).

- **Explanation:** The in vitro findings further support the in vivo results, indicating that fibulin-4 plays a direct role in cardiomyocyte function. The reduced beating frequency and contraction force in fibulin-4R/R microtissues suggest that fibulin-4 is necessary for maintaining the contractile machinery of cardiomyocytes. The lack of change in calcium sensitivity suggests that the impairment is not due to alterations in calcium handling but rather to intrinsic defects in the contractile apparatus, likely related to the compromised ECM. This highlights the importance of fibulin-4 in the structural and functional integrity of cardiomyocytes.

3. Impact of Fibulin-4 Reduction on Cardiac Response to Pressure Overload:

- **Observation:** When subjected to pressure overload via transverse aortic constriction (TAC), fibulin-4+/R mice (with a 50% reduction in fibulin-4 expression) showed markedly aggravated cardiac remodeling, dysfunction, and increased mortality compared to wild-type controls. This was evidenced by increased LV dilation, reduced EF, and elevated markers of hypertrophy and fibrosis.
- **Explanation:** The exacerbated response to TAC in fibulin-4+/R mice underscores the protective role of fibulin-4 in the heart's ability to adapt to increased mechanical stress. The reduced expression of fibulin-4 likely weakens the ECM, making the myocardium more susceptible to stress-induced damage. This leads to excessive remodelling, fibrosis, and eventually heart failure. The increase in mortality reflects the severity of the condition, suggesting that even partial reduction in fibulin-4 expression significantly compromises cardiac resilience under pathological conditions.

4. Gene Expression and Fibrotic Response:

- **Observation:** The study showed that fibulin-4 deficiency led to increased expression of hypertrophic and fibrotic markers, such as ANP, BNP, α SKA, and CTGF, following TAC. There was also increased collagen deposition and cardiomyocyte hypertrophy, particularly in fibulin-4+/R mice.
- **Explanation:** The upregulation of hypertrophic and fibrotic markers in fibulin-4-deficient mice following TAC is indicative of maladaptive cardiac remodeling. The increased fibrosis and cardiomyocyte hypertrophy suggest that the reduced fibulin-4 expression disrupts ECM signaling pathways, such as those mediated by TGF- β , leading to a profibrotic environment. This exacerbates the structural and functional deterioration of the heart, contributing to the observed cardiac dysfunction.

Conclusion: The detailed simulation results provided in this study clearly demonstrate that reduced fibulin-4 expression has profound effects on cardiac structure and function, both under normal conditions and in response to pressure overload. The findings underscore the importance of fibulin-4 in maintaining ECM integrity, cardiomyocyte function, and the heart's ability to adapt to mechanical stress. The insights gained from this study have significant implications for understanding the pathogenesis of fibulin-4-related cardiac

diseases and could inform the development of targeted therapies to mitigate the effects of fibulin-4 deficiency.

We now refer to this in the discussion section under working hypothesis (issue raised by editor and reviewer 1).